# TASK ARITHMETIC IN TRUST REGION: A TRAINING-FREE MODEL MERGING APPROACH TO NAVIGATE KNOWLEDGE CONFLICTS

## ABSTRACT

Multi-task model merging offers an efficient solution for integrating knowledge from multiple fine-tuned models, mitigating the significant computational and storage demands associated with multi-task training. As a key technique in this field, Task Arithmetic (TA) defines task vectors by subtracting the pre-trained model ($\theta_{\text{pre}}$) from the fine-tuned task models in parameter space, then adjusting the weight between these task vectors and $\theta_{\text{pre}}$ to balance task-generalized and task-specific knowledge. Despite the promising performance of TA, conflicts can arise among the task vectors, particularly when different tasks require distinct model adaptations. In this paper, we formally define this issue as **knowledge conflicts**, characterized by the performance degradation of one task after merging with a model fine-tuned for another task. Through in-depth analysis, we show that these conflicts stem primarily from the components of task vectors that align with the gradient of task-specific losses at $\theta_{\text{pre}}$. To address this, we propose **Task Arithmetic in Trust Region (TATR)**, which defines the trust region as dimensions in the model parameter space that cause only small changes (corresponding to the task vector components with gradient orthogonal direction) in the task-specific losses. Restricting parameter merging within this trust region, TATR can effectively alleviate knowledge conflicts. Moreover, TATR serves as both an independent approach and a plug-and-play module compatible with a wide range of TA-based methods. Extensive empirical evaluations on eight distinct datasets robustly demonstrate that TATR improves the multi-task performance of several TA-based model merging methods by an observable margin.

## 1 INTRODUCTION

The growing adoption of large foundation models is accompanied by significant practical challenges in terms of computational and storage demands (Kaplan et al., 2020). To address these challenges, multi-task model merging (Matena & Raffel, 2022) has emerged as a promising solution. For example, Task Arithmetic (Ilharco et al., 2023b) merges models by summing the task vectors from multiple tasks and applying them to the pre-trained model. Here task vectors are the difference in model parameters between the pre-trained foundation model and its fine-tuned version on a specific task. This approach builds a high-performance multi-task model by simple arithmetic operations in the model parameter space, thereby reducing computational overheads associated with fine-tuning on multiple tasks.

Despite their successes, task arithmetic and its variants (Yadav et al., 2023; Wang et al., 2024; Yang et al., 2024b;a) still suffer from conflicts between task vectors. As illustrated in Figure 1, adding task vectors pointing to

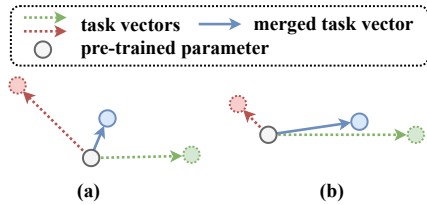

Figure 1: Illustration of knowledge conflicts between task vectors. In scenario (a), the two task vectors contain large-magnitude components in opposite directions. In scenario (b), the difference in vector magnitudes causes the merged model to be dominated by one task. Both lead to suboptimal performance in one or more tasks.

largely opposite directions may lead to catastrophic forgetting, and inconsistent task vector magnitudes may cause unbalanced merging, allowing the resulting model to be disproportionately influenced by a small subset of tasks. We refer to this issue as **knowledge conflicts**, represented as the expected performance variation of one task observed before and after merging another task vector. Knowledge conflicts differ from the typical notion of negative transfer (Yang et al., 2022; Meng et al., 2021; Liu et al., 2021b; Wang et al., 2023), as the former specifically refers to conflicts between predetermined, static task vectors, whereas the latter typically describes dynamic interference among tasks during training. Although current methods like sign alignment and test-time adaptation partially address knowledge conflicts, a thorough analysis of the root causes and a dedicated solution remain elusive.

In this paper, we propose a novel trust-region criterion for model merging, **Task Arithmetic in the Trust Region (TATR)**, which addresses the knowledge conflict problem. The trust region contains dimensions in the model parameter space that cause only small changes in the task-specific losses. When merging models, only the components of task vectors in the trust region are added to the pre-trained weights; other dimensions are discarded. TATR can be used independently or jointly with other techniques like Ties-Merging (Yadav et al., 2023), AdaMerging (Yang et al., 2024b), and Surgery (Yang et al., 2024a).

TATR utilizes the first-order Taylor series to compute the changes in the task-specific losses. It contrasts with a simplistic approach that selects components of task vectors that align with the negative gradient direction. While the simplistic approach is intuitive, empirical evidence reveals that it usually does not alleviate knowledge conflicts. We contend that, as the task vectors have large magnitudes, the first-order Taylor series fails to approximate the function well, leading to performance degradation. In contrast, TATR identifies directions that are orthogonal to the gradient, along which minimal cross-task interference happens. Due to overparameterization and parameter redundancy in the models (Dalvi et al., 2020; Chen et al., 2022b), such directions are usually not difficult to find.

In summary, the contributions of this paper are as follows:

- We conduct an analysis of knowledge conflicts that arise during model merging. Our investigation reveals that the components of task vectors aligned with the gradient of task-specific losses are the primary source of knowledge conflicts.
- We propose an approach, Task Arithmetic in the Trust Region (TATR), which defines a trust region to address knowledge conflicts. TATR serves as both an independent approach and a plug-and-play module compatible with a wide range of TA-based methods.
- We evaluate TATR through experiments across eight datasets. The experimental results demonstrate that TATR effectively mitigates knowledge conflicts and improves the performance of several TA-based model merging methods by an observable margin.

## 2 RELATED WORK

### 2.1 TRADITIONAL MULTI-TASK LEARNING

Multi-task learning (MTL) aims to improve performance by sharing knowledge across related tasks (Zhang & Yang, 2022). A significant challenge for MTL is negative transfer (Liu et al., 2017; Zhang et al., 2023b), where joint training on conflicting tasks yields performance lower than training on the tasks individually. Various solutions to negative transfer have been proposed, such as modularization (Tang et al., 2020; Ma et al., 2018), sparsification (Ding et al., 2021; Sun et al., 2020; Liu et al., 2019), and soft parameter sharing (Gao et al., 2020; Hazimeh et al., 2021). Other strategies focus on optimizing task interactions, such as adjusting task-specific loss weights (Sener & Koltun, 2018; Liu et al., 2019; 2022; Hu et al., 2023; Chen et al., 2022a), resolving gradient direction conflicts (Yu et al., 2020; Chen et al., 2020; Liu et al., 2021a; Javaloy & Valera, 2022; Navon et al., 2022), or preventing the dominance of certain tasks (Chen et al., 2018; He et al., 2022; Yang et al., 2023).

Traditional MTL are not well-suited for merging foundation models. First, retraining these models using vast amounts of data incurs significant computational costs. Large-scale foundation models are already resource-intensive, and training them with multi-task objectives further amplifies these demands, requiring immense computation and time. Additionally, retraining from scratch wastes

valuable knowledge optimized in each individual expert model. These considerations have driven the development of model merging as an alternative to multi-task learning.

## 2.2 MULTI-TASK LEARNING THROUGH MODEL MERGING

Model merging techniques, which aim to integrate knowledge across models, have attracted increasing attention in recent years. As a precursor, Stochastic Weight Averaging (SWA) (Izmailov et al., 2018) averages model weights near the end of training. This concept was further advanced by approaches like SWAD (Cha et al., 2021) and Ensemble of Averages (EoA) (Arpit et al., 2022). Empirical evidence from Ilharco et al. (2023a) demonstrates that parameter averaging effectively integrates knowledge from models trained on diverse tasks. DLCPA (Sun et al., 2023) proposes to apply cumulative parameter averaging (CPA) to continually assimilate knowledge across distinct tasks. Fisher-Merging (Matena & Raffel, 2022) leverages the Fisher information matrix Fisher (1925) to measure the importance of model parameters and merge models using weighted averaging. Additionally, RegMean (Jin et al., 2023) formulates an optimal merging model by minimizing the distance to each model in the parameter space.

Recently, Task Arithmetic (TA) (Ilharco et al., 2023b) innovatively proposes the concept of "task vector", defined as the vector from a pre-trained model to its fine-tuned counterpart in the parameter space. By weighting these task vectors and adding them back to the pre-trained model, TA strikes a harmonious balance between generalized knowledge from the pre-train model and the task-specific knowledge in the task vectors. Following this insight, Ties-Merging (Yadav et al., 2023) refines the fusion process by discarding parameters deemed insignificant or of low magnitude. PEFT (Zhang et al., 2023a) and MoLE (Wu et al., 2024) further extend TA by integrating it with LoRA (Hu et al., 2022) modules. Furthermore, Ortiz-Jimenez et al. (2023) suggests fine-tuning models in the tangent space, which can effectively mitigate conflict between task vectors.

Furthermore, several approaches combine test-time adaptation techniques with TA, yielding superior MTL performance. These test-time adaptation-based methods typically allocate merging weights and fine-tune them during testing using unsupervised test data. For instance, AdaMerging (Yang et al., 2024b) trains a set of merging coefficients for layers, while other methods fit lightweight adapter modules, such as representation surgery (Yang et al., 2024a) and MoE router (Tang et al., 2024).

## 3 PRELIMINARIES

### 3.1 PROBLEM SETTING

Formally, let $\theta_{\text{pre}} \in \mathbb{R}^N$ denote the set of $N$ parameters of a pre-trained model, which is initially trained using a diverse, large-scale dataset to encapsulate generalized, task-agnostic knowledge. Subsequently, $\theta_{\text{pre}}$ undergoes fine-tuning for $K$ distinct downstream tasks, yielding a set of fine-tuned parameters $\{\theta_k\}_{k=1}^K$, where each $\theta_k$ is tailored to a specific task $k$.

The objective of model merging is to integrate these fine-tuned parameters from the task-specific models $\{\theta_k\}_{k=1}^K$ into a single model $\theta_{\text{MTL}}$. This merged model $\theta_{\text{MTL}}$ aims to achieve effective generalization across all $K$ tasks without resorting to trivial solutions such as retraining from scratch or requiring full access to the training datasets of all tasks.

### 3.2 TASK ARITHMETIC

Task arithmetic (TA) (Ilharco et al., 2023b) is known as a competitive baseline for model merging by leveraging **task vectors**, which are defined as the differential parameters between the pre-trained model $\theta_{\text{pre}}$ and each fine-tuned model. Specifically, the task vector for task $k$ is given by:

$$\Delta_k = \theta_k - \theta_{\text{pre}}. \tag{1}$$

TA posits that these task vectors encapsulate essential task-specific knowledge. The merged model is then constructed by adding the cumulative task vector from all tasks back to $\theta_{\text{pre}}$:

$$\theta_{\text{TA}} = \underbrace{\theta_{\text{pre}}}_{\text{task-generalized}} + \lambda \underbrace{\sum_k \Delta_k}_{\text{task-specific}} , \tag{2}$$

where $\lambda > 0$ is a pre-defined hyper-parameter that governs the influence of task-specific adjustments.

This method is favored over direct averaging of fine-tuned parameters as it seeks a balance between generalized and task-specific knowledge, contributing to its competitive advantage. Nevertheless, task vectors can encode conflicting adaptations across different tasks, leading to potential knowledge conflicts that may result in information loss and diminished performance. This issue, termed as "knowledge conflict", will be dissected further in the subsequent section.

## 4 ANALYZING KNOWLEDGE CONFLICT

Knowledge conflict frequently arises when merging MTL models, as the expert models encapsulate diverse, sometimes conflicting, knowledge. We formally define knowledge conflict as follows:

**Definition 1** (Knowledge Conflict)**.** *Given a pre-trained model $\theta_{\text{pre}}$ and a set of fine-tuned, task-specific models $\{\theta_k\}_{k=1}^K$, where $\theta_k$ represents the parameters optimized for task $k$, the knowledge conflict on task $j$ caused by task $i$ can be quantified by the change in performance of task $j$ when task $i$ is included in the model merging process. Formally, the knowledge conflict is defined as*

$$\mathcal{C}_{j|i} := L_j \left( \theta_{\text{MTL}}(\{\theta_k\}_{k=1}^K) \right) - L_j \left( \theta_{\text{MTL}}(\{\theta_k\}_{k \neq i}) \right),$$

*where $L_j(\theta)$ denotes the loss for task $j$ with model parameters $\theta$, and $\theta_{\text{MTL}}(\{\theta_k\}_{k \neq i})$ represents the merged model parameters excluding the model fine-tuned for task $i$. The overall knowledge conflict, $\mathcal{C}$, is computed as the sum of $\mathcal{C}_{j|i}$ across all task pairs $(i, j)$:*

$$\mathcal{C} := \sum_{i \neq j} \mathcal{C}_{j|i}.$$

*A **higher** value of $\mathcal{C}$ indicates a greater degree of conflict, as it reflects a larger negative impact on task $j$'s performance when task $i$ is incorporated into the merging process.*

Knowledge conflict can be regarded as a special case of negative transfer, although these concepts emphasize different aspects. In traditional MTL, negative transfer typically refers to the *dynamic* interference between tasks during joint training, where conflicting gradients impede the model from learning effective representations (Zhang et al., 2023b). In contrast, the knowledge conflict defined here highlights a *static* nature among the fine-tuned model parameters, where further training to resolve task interference is prohibited. Each fine-tuned model has already encoded task-specific knowledge, which may be inherently incompatible with that of other tasks. As a result, knowledge conflict in model merging presents a unique challenge, necessitating methods that can align or reconcile parameters without resorting to retraining.

In the context of TA, knowledge conflict can be further articulated through task vectors:

$$\mathcal{C}_{\text{TA}j|i} = L_j \left( \theta_{\text{pre}} + \lambda \sum_k \Delta_k \right) - L_j \left( \theta_{\text{pre}} + \lambda \sum_{k \neq i} \Delta_k \right). \tag{3}$$

An intuitive hypothesis is that task vector components aligned with the gradient ascent direction contribute to knowledge conflicts. More formally, we apply a Taylor expansion around $\theta_{\text{pre}}$ on the

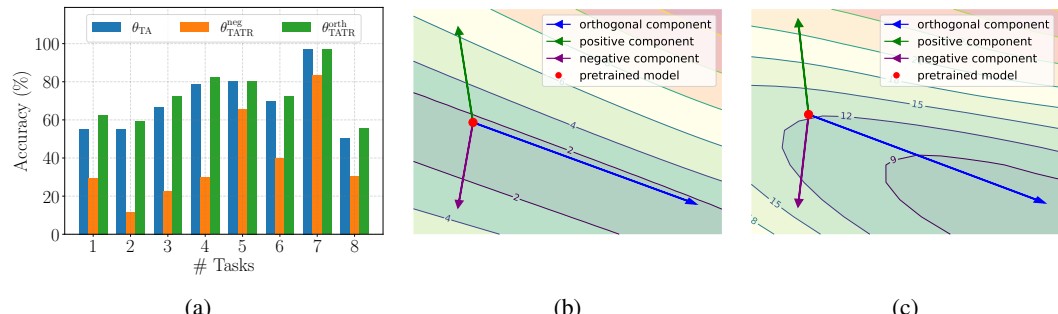

(a)          (b)          (c)

Figure 2: (a) Performance comparison across eight datasets (Cf. Section 6.1) when merging negative components (e.g., components aligned with the loss descent direction) and orthogonal components (e.g., components orthogonal to the gradient) of task vectors, corresponding to $\theta_{\text{TATR}}^{\text{neg}}$ and $\theta_{\text{TATR}}^{\text{orth}}$, respectively. (b) Loss landscape of the EuroSAT dataset and the components of the cumulative task vector from the remaining seven datasets. (c) The total loss landscape over all eight datasets, along with the components of the cumulative across task vectors. Note that in both (b) and (c), the loss landscape is visualized in a hyperplane going through the three points: $\theta_{pre} + \Delta^{\perp}$, $\theta_{pre} + \Delta^{+}$, and $\theta_{pre} + \Delta^{-}$. Refer Section B in Appendix for more details.

right-hand side of Eq. (3):

$$
L_j\left(\theta_{\text{pre}} + \lambda \sum_k \Delta_k\right) - L_j\left(\theta_{\text{pre}} + \lambda \sum_{k \neq i} \Delta_k\right)
$$

$$
\approx L_j\left(\theta_{\text{pre}}\right) + \left\langle \nabla_\theta L_j\left(\theta_{\text{pre}}\right), \lambda \sum_k \Delta_k \right\rangle - L_j\left(\theta_{\text{pre}}\right) - \left\langle \nabla_\theta L_j\left(\theta_{\text{pre}}\right), \lambda \sum_{k \neq i} \Delta_k \right\rangle \tag{4}
$$

$$
= \lambda \left\langle \nabla_\theta L_j\left(\theta_{\text{pre}}\right), \Delta_i \right\rangle = \lambda \sum_{n=1}^{N} \nabla_\theta L_j\left(\theta_{\text{pre}}\right)[n] \cdot \Delta_i[n].
$$

where $v[n]$ selects the $n$-th component of the vector $v$. Equation (4) suggests that task vector components aligned with the gradient **ascent** direction are primarily responsible for knowledge conflicts, while those in line with the gradient descent direction should be prioritized during model merging. That is, we should avoid merging the $n$-th component of the task vector $\Delta_i$ if $\nabla_\theta L_j\left(\theta_{\text{pre}}\right)[n] \cdot \Delta_i[n]$ is large.

On the other hand, perhaps counter-intuitively, empirical evidence (Figure 2(a)) shows that merging the gradient **descent** components (i.e., $\nabla_\theta L_j\left(\theta_{\text{pre}}\right)[n] \cdot \Delta_i[n] < 0$) causes a significant performance drop. A potential explanation for this phenomenon is that the task vectors have large magnitudes, thereby the first-order Taylor expansion cannot offer a good approximation of the task loss $L_j$. As a result, even if we merge a component in the gradient descent direction, we can overshoot the local optimum and end up increasing the task loss (Ruder, 2017).

To facilitate analysis, we decompose the task vector $\Delta_i$ into three components:

- **Orthogonal component**, which contains elements with near-zero ~~inner~~ product $\Delta_i^{\perp} = \Delta_i \odot \mathbb{1}_{\{\nabla_\theta L_j(\theta_{\text{pre}}) \odot \Delta_i \approx 0\}}$;

- **Positive component**, with elements having a positive ~~inner~~ product $\Delta_i^{+} = \Delta_i \odot \mathbb{1}_{\{\nabla_\theta L_j(\theta_{\text{pre}}) \odot \Delta_i > 0\}}$;

- **Negative component**, defined by elements with a negative ~~inner~~ product $\Delta_i^{-} = \Delta_i \odot \mathbb{1}_{\{\nabla_\theta L_j(\theta_{\text{pre}}) \odot \Delta_i < 0\}}$.

Here, $\odot$ denotes the Hadamard (element-wise) product, and $\mathbb{1}_{\{p\}} \in \mathbb{R}^N$ is an indicator vector that takes the value 1 in the dimension that $p$ is true and 0 otherwise.

We illustrate the impact of the three components within the loss landscape in Figure 2 (b). It is evident that the positive component leads to an increase in loss, as the model moves in the gradient

ascent direction. The orthogonal component results in relatively smooth changes in the loss. **Interestingly, while the negative component initially follows the descent direction of the loss, it overshoots the local optimum, ultimately leading to an increase in loss.** As a result, the total loss across all tasks shown in Figure 2 (c) highlights that the orthogonal component is more beneficial for knowledge fusion than either the positive or negative components.

## 5 TASK ARITHMETIC IN THE TRUST REGION

The above observations are reasonable since neural network parameters, particularly those in pre-trained foundation models, often exhibit high redundancy (Dalvi et al., 2020; Chen et al., 2022b). Additionally, task-specific knowledge is often low-rank Hu et al. (2022), i.e., only a few parameter directions are critical for learning the task. In order to identify a small set of critical parameters that should not be altered during model merging and alleviate knowledge conflicts, we propose defining the following trust region:

**Definition 2** (Trust Region for Knowledge Conflict). *Given a pre-trained model $\theta_{\text{pre}}$, the trust region specific in the **dimension space** is defined as follows:*

$$\mathcal{TR} := \left\{ n \middle| \sum_{i \neq j} \left| \nabla_\theta L_j(\theta_{\text{pre}})[n] \cdot \Delta_i[n] \right| < \epsilon \right\}, \tag{5}$$

*where $n \leq N$ indexes the dimensions of the parameter space, $\epsilon$ represents the sensitivity threshold, and any dimension exceeding this threshold will be excluded from the trust region and not permitted to merge.*

Dimensions outside the trust region (corresponding components of task vector that are collinear with the gradient direction, regardless of whether the directions are aligned or opposite) are likely to cause knowledge conflicts. Conversely, when $\nabla_\theta L_j(\theta_{\text{pre}})$ and $\Delta_i$ are orthogonal, their projections minimally interfere with each other, thereby reducing knowledge conflict.

We are now ready to present the TATR method. TATR mitigates knowledge conflict by restricting merging within the trust region, involving the following three key steps.

**Calculating task-specific gradients.** The first step involves computing the gradient for each task. Since accessing the full training data for each task is often impractical, we approximate the gradient using an exemplar set for each task, denoted as $\{S_1, \ldots, S_K\}$. For each task, the absolute gradient of the loss function $L_k(.)$ (cross-entropy loss in our experiments) is computed as follows:

$$\left| \nabla_\theta L_k(\theta_{\text{pre}}) \right| \approx \mathbb{E}_{x_k \in S_k} \left| \nabla_\theta L_k(x_k; \theta_{\text{pre}}) \right|. \tag{6}$$

Notably, we place the expectation outside the absolute value operation, drawing inspiration from the Fisher Information Matrix (Wasserman, 2013). This design captures absolute gradients that reflect the average variation of parameters, facilitating the measurement of knowledge conflict across every exemplar. Additionally, the exemplar size can be remarkably small. Our empirical results in Figure 3 (a) show that even in a one-shot setting, we achieve a competitive average accuracy of 72.3%, which is close to the highest accuracy of 72.8% obtained with 16 samples. Similar results are observed when TATR is integrated into AdaMerging Yang et al. (2024b).

We also propose a zero-shot version, where the task vector is used to estimate the gradient. Although there may be estimation errors, this approach still offers performance improvements for TA-based methods in most scenarios:

$$\left| \nabla_\theta L_k(\theta_{\text{pre}}) \right| \approx |\Delta_k|. \tag{7}$$

**Establishing the trust region.** Next, we aim to identify the trust region with minimal knowledge conflict, with a key requirement being the determination of the sensitivity threshold $\epsilon$. However, manually specifying the exact value of $\epsilon$ becomes complex and tedious. Therefore, we employ a ranking method to infer $\epsilon$. To achieve this, we derive the sensitivity of each dimension that may cause knowledge conflict, based on Definition 2:

$$\Omega^{\text{Trust}} = \sum_{i \neq j} \left| \nabla_\theta L_j(\theta_{\text{pre}}) \odot \Delta_i \right| = \sum_{i \neq j} \left| \nabla_\theta L_j(\theta_{\text{pre}}) \right| \odot |\Delta_i|. \tag{8}$$

---

**Algorithm 1:** The model merging process of TATR

**Input:** Pre-trained model $\theta_{\text{pre}}$; Task vectors $\{\Delta_1, \ldots, \Delta_K\}$; Exemplar-set $\{S_1, \ldots, S_K\}$
**Output:** Merged model $\theta_{\text{TATR}}$

1 **// Deriving gradients for each task**
2 **for** $k = 1, \ldots, K$ **do**
3 $\quad \lfloor \ G_k = \mathbb{E}_{x_k \in S_k} |\nabla_\theta L_k (x_k; \theta_{\text{pre}})|$
4 **// Establishing the trust region**
5 $\Omega^{\text{Trust}} = \sum_{i \neq j} G_j \odot |\Delta_i|$
6 $\epsilon = \text{proportion\_selection}(\Omega^{\text{Trust}}, \tau)$
7 $\mathcal{TR} = \{n \mid \Omega^{\text{Trust}}[n] < \epsilon\}$
8 **// Merging**
9 $\theta_{\text{TATR}} = \theta_{\text{pre}} + \lambda \sum_k \Delta_k \odot \mathbb{1}_{\{n \in \mathcal{TR}\}}$
10 **return** $\theta_{\text{TATR}}$

---

Next, the sensitivity threshold $\epsilon$ of the trust region is determined through a proportional selection operation:

$$\epsilon = \text{proportion\_selection}(\Omega^{\text{Trust}}, \tau). \tag{9}$$

In this process, $\Omega^{\text{Trust}}$ is sorted in descending order, and the values corresponding to the predefined ratio $\tau$ are selected as the sensitivity threshold $\epsilon$. Based on $\epsilon$, we are able to establish the trust region $\mathcal{TR}$ according to Definition 2.

**Merging the task vectors.** The final step involves merging the task vectors using TA, where the merging occurs within the dimensions confined to the trust region:

$$\theta_{\text{TATR}} = \theta_{\text{pre}} + \lambda \sum_k \Delta_k \odot \mathbb{1}_{\{n \in \mathcal{TR}\}}, \tag{10}$$

where $\mathbb{1}_{\{n \in \mathcal{TR}\}} \in \mathbb{R}^N$ is an indicator vector whose value is 1 at index $n$ if $n$ belongs to the trust region and 0 otherwise. The detailed workings of TATR are outlined in Algorithm 1. The entire merging process does not rely on any additional training process.

Moreover, the techniques introduced in TATR selectively limit the merging process to a subset of model parameters, allowing it to function as a plug-and-play module that seamlessly integrates with a wide range of TA-based approaches, such as:

- **Ties-Merging & TATR:** Ties-Merging (Yadav et al., 2023) partially reduces knowledge conflicts by pruning low-magnitude parameters and aligning the signs of task vectors. However, this approach overlooks conflicts that may arise from high-magnitude parameters. This bias can lead to knowledge conflicts, where some tasks dominate the model's behavior. The combination of TATR with Ties-Merging refines the process, as shown in the following formula:

$$\theta_{\text{Ties+TATR}} = \theta_{\text{pre}} + \lambda \sum_k \Phi(\Delta_k) \odot \mathbb{1}_{\{n \in \mathcal{TR}\}}, \tag{11}$$

  where $\Phi(.)$ indicates the TrIm, Elect Sign, and Merge operation of Ties-Merging.

- **AdaMerging & TATR:** AdaMerging (Yang et al., 2024b) adaptively learns merging coefficients but does not inherently resolve knowledge conflicts between task vectors. This can lead to interference during coefficient learning, especially when tasks require opposing parameter adaptations. TATR addresses this by pre-filtering task vectors to retain only those components within the trust region, ensuring that AdaMerging operates in a conflict-reduced parameter space:

$$\theta_{\text{Ada+TATR}} = \theta_{\text{pre}} + \sum_k \lambda_k \Delta_k \odot \mathbb{1}_{\{n \in \mathcal{TR}\}}, \tag{12}$$

  where $\lambda_1, \ldots, \lambda_K$ represent the learnable coefficients for AdaMerging.

- **Surgery & TATR:** Similarly, Surgery (Yang et al., 2024a) introduces additional modules to align task-specific features during merging. TATR complements Surgery by pre-selecting components of task vectors that reside in the trust region. The integrated approach is formalized as:

$$\theta_{\text{Surgery+TATR}} = \left\{ \theta_{\text{surgery}}, \theta_{\text{pre}} + \lambda \sum_k \Delta_k \odot \mathbb{1}_{\{n \in \mathcal{TR}\}} \right\}, \quad (13)$$

where $\theta_{\text{surgery}}$ denotes the additional parameters introduce by the Surgery module.

# 6 EXPERIMENTS

## 6.1 SETTINGS

**Datasets.** Following prior works (Ilharco et al., 2023b; Yadav et al., 2023; Yang et al., 2024b;a), we perform model merging on the following eight datasets: SUN397 (Xiao et al., 2016), Cars (Krause et al., 2013), RESISC45 (Cheng et al., 2017), EuroSAT (Helber et al., 2019), SVHN (Netzer et al., 2011), GTSRB (Stallkamp et al., 2011), MNIST (LeCun & Cortes, 2010), DTD (Cimpoi et al., 2014).

**Baselines.** We compare our approach against a diverse set of methods, categorized into basic baseline methods, test-time training-based model merging methods, and training-free model merging methods. Basic baseline methods include the Pre-trained model, Individual task model, and the Traditional Multi-Task Learning model. For test-time training-based methods, we provide AdaMerging, AdaMerging++ (Yang et al., 2024b), and Surgery (Yang et al., 2024a). Among the training-free methods, we consider the simple Weight Average, Fisher Merging (Matena & Raffel, 2022), RegMean (Jin et al., 2023), Task Arithmetic (Ilharco et al., 2023b), and Ties-Merging (Yadav et al., 2023).

**Implementation details.** Our implementation strictly follows task arithmetic (Ilharco et al., 2023b) and AdaMerging (Yang et al., 2024b). We apply the ViT-B/32 and ViT-L/14 in CLIP (Radford et al., 2021) as the pre-trained model. Task vectors are derived from task arithmetic (Ilharco et al., 2023b) which is fine-tuned on each specific dataset. We report the accuracy of each task after merging the models, along with the average accuracy (i.e., Avg ACC). The hyper-parameter $\tau$ is tuned within the range $[0.1\%, 0.2\%, 0.5\%, 1.0\%, 2.0\%, 5.0\%]$, while the size of the exemplar set is fixed at 128. Additional implementation details can be found in our supplementary code.

## 6.2 PERFORMANCE COMPARISON

The performance of all baselines using the ViT-B/32 and ViT-L/14 architectures is presented in Table 1 and Table 2, respectively. We report the performance metrics for each task after merging, as well as the overall average performance.

As illustrated in the tables, the pre-trained model exhibits the lowest performance across all methods, due to the absence of task-specific supervision. In contrast, the Individual models achieve the highest performance, as they are exclusively trained for each specific task, which thus represents the upper-bound performance for model merging. Traditional MTL encounters knowledge conflict issues, resulting in slightly lower performance compared to the Individual models.

Among the model merging methods, the simplest Weight Averaging suffers significant knowledge conflicts, resulting in a worse performance. Fisher Merging and RegMean improve Weight Averaging by incorporating parameter importance weight into the averaging process. TA and its enhanced version, Ties-merging, demonstrate substantial performance improvements by better balancing the pre-trained and task-specific knowledge. Additionally, owing to the additional training process, test-time training-based models (AdaMerging and Surgery) generally outperform the training-free methods.

Our proposed TATR method belongs to the training-free model merging approach. As the techniques of TATR are orthogonal to existing model merging methods, we also report performance when TATR is plugged into strong baselines. The experimental results demonstrate that TATR consistently enhances all TA-based methods. When incorporated into task arithmetic, both TATR and its zero-shot version lead to significant performance improvements, increasing average accuracy by 3.7%

and 1.5% on ViT-B/32, and by 0.8% and 0.1% on ViT-L/14, respectively. The best results are obtained when TATR is combined with layer-wise AdaMerging++, achieving an average accuracy of 82.5% on ViT-B/32 and 91.5% on ViT-L/14.

Table 1: Multi-task performance when merging ViT-B/32 models on eight tasks. The column of "# Best" indicates the number of datasets on which the proposed method achieved the best performance, and the best and second-best performance are highlighted with **bold** and underline.

| Method | SUN397 | Cars | RESISC45 | EuroSAT | SVHN | GTSRB | MNIST | DTD | # Best | Avg Acc |
|---|---|---|---|---|---|---|---|---|---|---|
| *Basic baseline methods* | | | | | | | | | | |
| Pre-trained | 62.3 | 59.7 | 60.7 | 45.5 | 31.4 | 32.6 | 48.5 | 43.8 | - | 48.0 |
| Individual | 75.3 | 77.7 | 96.1 | 99.7 | 97.5 | 98.7 | 99.7 | 79.4 | - | 90.5 |
| Traditional MTL | 73.9 | 74.4 | 93.9 | 98.2 | 95.8 | 98.9 | 99.5 | 77.9 | - | 88.9 |
| *Test-time training based methods* | | | | | | | | | | |
| TW AdaMerging | 58.0 | 53.2 | 68.8 | 85.7 | 81.1 | 84.4 | 92.4 | 44.8 | 0 | 71.1 |
| TW AdaMerging++ | 60.8 | 56.9 | 73.1 | 83.4 | 87.3 | 82.4 | 95.7 | 50.1 | 0 | 73.7 |
| LW AdaMerging | 64.5 | 68.1 | 79.2 | 93.8 | 87.0 | **91.9** | 97.5 | 59.1 | 1 | 80.1 |
| LW AdaMerging++ | 66.6 | 68.3 | 82.2 | 94.2 | 89.6 | 89.0 | 98.3 | 60.6 | 0 | 81.1 |
| Surgery Merging | 63.8 | 59.9 | 83.3 | **97.9** | 87.0 | 87.0 | 98.6 | 69.4 | 1 | 80.9 |
| **LW AdaMerging++ & TATR zero-shot (Ours)** | **72.0** | **70.8** | 81.5 | 88.9 | 84.9 | 84.2 | **99.3** | 66.7 | 3 | 81.0 |
| **LW AdaMerging++ & TATR (Ours)** | 69.8 | 70.3 | 83.7 | 93.7 | **90.0** | 90.2 | 98.3 | 63.7 | 1 | **82.5** |
| **Surgery & TATR zero-shot (Ours)** | 64.2 | 60.4 | 82.7 | 96.9 | 86.4 | 86.5 | 98.5 | 68.7 | 0 | 80.5 |
| **Surgery & TATR (Ours)** | 67.1 | 62.2 | **87.1** | 97.4 | 87.3 | 88.5 | 98.7 | 70.9 | 2 | 82.4 |
| *Training-free methods* | | | | | | | | | | |
| Weight Averaging | 65.3 | 63.4 | 71.4 | 71.7 | 64.2 | 52.8 | 87.5 | 50.1 | 0 | 65.8 |
| Fisher Merging | **68.6** | **69.2** | 70.7 | 66.4 | 72.9 | 51.1 | 87.9 | **59.9** | 3 | 68.3 |
| RegMean | 65.3 | 63.5 | 75.6 | 78.6 | 78.1 | 67.4 | 93.7 | 52.0 | 0 | 71.8 |
| Task Arithmetic | 55.2 | 54.9 | 66.7 | 78.9 | 80.2 | 69.7 | 97.3 | 50.4 | 0 | 69.1 |
| Ties-Merging | 59.8 | 58.6 | 70.7 | 79.7 | **86.2** | 72.1 | 98.3 | 54.2 | 2 | 72.4 |
| **TATR zero-shot (Ours)** | 59.0 | 56.6 | 69.2 | 80.2 | 79.0 | 70.5 | 97.0 | 53.5 | 0 | 70.6 |
| **TATR (Ours)** | 62.7 | 59.3 | 72.3 | **82.3** | 80.5 | **72.6** | 97.0 | 55.4 | 1 | 72.8 |
| **Ties-Merging & TATR zero-shot (Ours)** | 64.9 | 64.2 | 74.7 | 76.4 | 81.2 | 69.3 | 96.5 | 54.3 | 1 | 72.7 |
| **Ties-Merging & TATR (Ours)** | 66.3 | 65.9 | **75.9** | 79.4 | 79.9 | 68.1 | 96.2 | 54.8 | 1 | **73.3** |

Table 2: Multi-task performance when merging ViT-L/14 models on eight tasks.

| Method | SUN397 | Cars | RESISC45 | EuroSAT | SVHN | GTSRB | MNIST | DTD | # Best | Avg Acc |
|---|---|---|---|---|---|---|---|---|---|---|
| *Basic baseline methods* | | | | | | | | | | |
| Pre-trained | 66.8 | 77.7 | 71.0 | 59.9 | 58.4 | 50.5 | 76.3 | 55.3 | - | 64.5 |
| Individual | 82.3 | 92.4 | 97.4 | 100.0 | 98.1 | 99.2 | 99.7 | 84.1 | - | 94.2 |
| Traditional MTL | 80.8 | 90.6 | 96.3 | 96.3 | 97.6 | 99.1 | 99.6 | 84.4 | - | 93.5 |
| *Test-time training based methods* | | | | | | | | | | |
| AdaMerging | 79.0 | 90.3 | 90.8 | 96.2 | **93.4** | **98.0** | 99.0 | 79.9 | 2 | 90.8 |
| AdaMerging++ | 79.4 | 90.3 | 91.6 | 97.4 | **93.4** | 97.5 | 99.0 | 79.2 | 1 | 91.0 |
| Surgery Merging | 75.7 | 84.4 | 93.1 | **98.8** | 91.3 | 93.4 | 99.1 | 76.1 | 1 | 89.0 |
| **AdaMerging & TATR zero-shot (Ours)** | 80.7 | **95.3** | 95.0 | 94.9 | 84.7 | 92.4 | **99.8** | 86.0 | 1 | 91.1 |
| **AdaMerging++ & TATR (Ours)** | **81.6** | **95.9** | **95.8** | 95.5 | 83.2 | 92.6 | 99.7 | **87.5** | 4 | **91.5** |
| **Surgery & TATR zero-shot (Ours)** | 75.6 | 85.1 | 93.8 | 98.5 | 91.0 | 93.1 | 99.2 | 76.3 | 0 | 89.1 |
| **Surgery & TATR (Ours)** | 76.3 | 85.8 | 93.8 | **98.8** | 91.4 | 93.0 | 99.2 | 77.9 | 1 | 89.5 |
| *Training-free methods* | | | | | | | | | | |
| Weight Averaging | 72.1 | 81.6 | 82.6 | 91.9 | 78.2 | 70.7 | 97.1 | 62.8 | 0 | 79.6 |
| Fisher Merging | 69.2 | **88.6** | 87.5 | 93.5 | 80.6 | 74.8 | 93.3 | 70.0 | 1 | 82.2 |
| RegMean | 73.3 | 81.8 | 86.1 | **97.0** | 88.0 | 84.2 | 98.5 | 60.8 | 1 | 83.7 |
| Task Arithmetic | 73.9 | 82.1 | 86.6 | 94.1 | 87.9 | 86.7 | 98.9 | 65.6 | 0 | 84.5 |
| Ties-Merging | **76.5** | 85.0 | **89.3** | 95.7 | 90.3 | 83.3 | 99.0 | **68.8** | 3 | 86.0 |
| **TATR zero-shot (Ours)** | 74.3 | 81.5 | 86.6 | 92.7 | 88.6 | 88.1 | 99.1 | 66.0 | 0 | 84.6 |
| **TATR (Ours)** | 74.6 | 83.7 | 87.6 | 93.7 | 88.6 | 88.1 | 99.0 | 66.8 | 0 | 85.3 |
| **Ties-Merging & TATR zero-shot (Ours)** | 75.8 | 85.3 | 89.2 | 94.7 | 89.1 | 87.1 | 99.0 | 68.6 | 0 | 86.1 |
| **Ties-Merging & TATR (Ours)** | 76.3 | 85.3 | 88.8 | 94.4 | **90.8** | **88.7** | **99.2** | **68.8** | 4 | 86.5 |

## 6.3 SENSITIVITY ANALYSIS OF HYPERPARAMETERS

This section presents an analysis of the model's sensitivity to two hyperparameters: the number of exemplar samples and the proportion $\tau$ in Eq. (9). As shown in Figure 3, the performance of TATR remains stable with respect to both hyperparameters. Furthermore, Figure 3 (a) demonstrates that even in a one-shot setting, TATR achieves a competitive average accuracy of 72.3%, evidently outperforming Task Arithmetic (0 exemplars in Figure 3 (a)) and comparable to the highest accuracy of 72.8%. Similarly, experiments plugged into AdaMerging also support this phenomenon, where the one-shot scenario achieves an average accuracy of 82.3%, nearly matching the peak accuracy of 82.5%. Additionally, Figure 3 (b) suggests that excluding a small proportion of parameters (less than 1%) is sufficient to alleviate the knowledge conflicts.

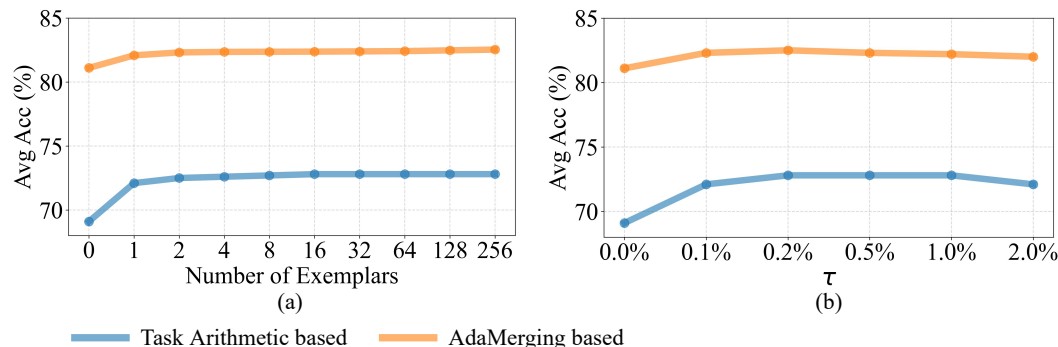

Figure 3: Average accuracy (%) of TATR on eight tasks versus the number of exemplars (a) and $\tau$ (b).

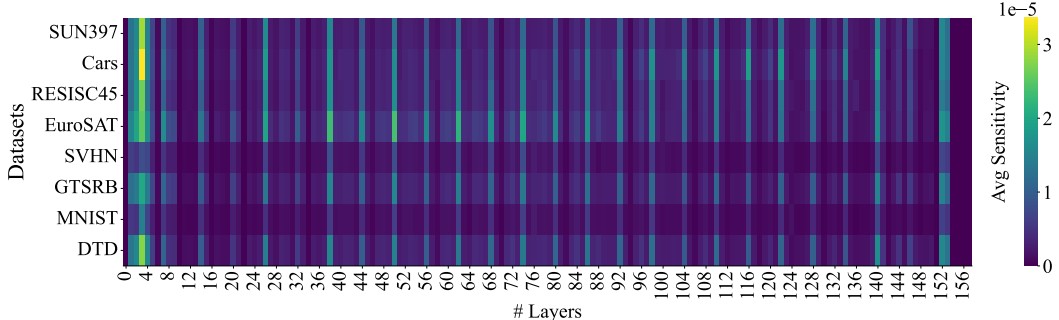

Figure 4: The average sensitivity of each dataset to task vectors across layers.

### 6.4 ANALYSIS OF SENSITIVITY $\Omega^{\mathrm{TRUST}}$ FOR KNOWLEDGE CONFLICT

Figure 4 illustrates the average sensitivity of each dataset to task vectors across different layers. Three key characteristics can be observed. Firstly, the shallow layers exhibit greater sensitivity than other layers. Shallow layers typically encode task-generalized knowledge, and the increased sensitivity highlights the importance of preserving this information in the TATR method. Secondly, the sensitivity exhibits periodic variations across layers, with bias layers generally exhibiting higher sensitivity than weight layers. This trend is reasonable, as bias layers have a more pronounced impact on network outputs, making them more susceptible to knowledge conflicts. Lastly, datasets comprising digit data (e.g., SVHN and MNIST) show relatively lower sensitivity to knowledge conflicts, which can be attributed to their significant domain differences from other real-world datasets.

## 7 CONCLUSION

In this paper, we delve deep into the critical challenge of knowledge conflict in multi-task model merging with a focus on task arithmetic. We began by formalizing the concept of knowledge conflict as the degradation in model performance caused by the interference between task vectors. Our analysis and empirical findings suggest that components of task vectors orthogonal to the gradient direction exhibit minimal knowledge conflict. This insight motivates us to define a trust region based on orthogonality and propose Task Arithmetic in the Trust Region (TATR). Extensive experiments across eight diverse datasets demonstrate that TATR effectively mitigates the knowledge conflict, enhancing the overall multi-task performance of task arithmetic-based methods.

## 8 REPRODUCIBILITY STATEMENT

We have included the complete source code in the supplementary materials, and will release the full codebase as open source upon publication. All datasets and settings are documented for clarity.

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

## A EXPERIMENT DETAILS

This section provides details of experiments, including the description of the experimental environment, datasets, and baselines.

## A.1 ENVIRONMENT

All experiments detailed in our manuscript and appendix were conducted on a workstation running Ubuntu 16.04, equipped with 18 Intel Xeon 2.60GHz CPUs, 256 GB of memory, and 6 NVIDIA RTX3090 GPUs. Python 3.8 was used to implement all the methods.

## A.2 DATASETS

Our experiments strictly follow Task Arithmetic (Ilharco et al., 2023b) and AdaMerging (Yang et al., 2024b), utilizing eight widely-used image classification datasets. The information of these datasets is described as follows:

- **SUN397** (Xiao et al., 2016): A scene classification dataset containing 108,754 images across 397 classes. Each class includes at least 100 images.

- **Stanford Cars (Cars)** (Krause et al., 2013): A car classification dataset featuring 16,185 images of 196 car categories. The dataset is evenly split between training and test sets.

- **RESISC45** (Cheng et al., 2017): A remote sensing image classification dataset comprising 31,500 images across 45 scene categories, with approximately 700 images per class.

- **EuroSAT** (Helber et al., 2019): A satellite image classification dataset consisting of 27,000 labeled and geo-referenced images distributed among 10 categories.

- **SVHN** (Netzer et al., 2011): A real-world digit classification dataset derived from house numbers in Google Street View images. It includes 10 classes, with a training set of 73,257 images, a test set of 26,032 images, and an additional 531,131 samples available for extended training.

- **GTSRB** (Stallkamp et al., 2011): A traffic sign classification dataset comprising more than 50,000 images across 43 traffic sign categories.

- **MNIST** (LeCun & Cortes, 2010): A well-known benchmark for handwritten digit classification, containing 60,000 training images and 10,000 test images, evenly distributed among 10 classes of digit numbers.

- **DTD** (Cimpoi et al., 2014): A texture classification dataset consisting of 5,640 images distributed across 47 texture classes, with approximately 120 images per class.

## A.3 BASELINES.

In our experiments, we compare our methods with several baseline approaches, which are grouped into four categories: basic baseline methods, test-time training-based methods, training-free methods, and our proposed methods. The details of these methods are as follows:

**i) Basic baseline methods:**

- **Pre-trained** directly employs a pre-trained model to predict across multiple tasks. Since it does not incorporate any downstream task-specific information during model training, its performance on downstream tasks is typically suboptimal.

- **Individual**. In this approach, an independent fine-tuned model is used for each task. While it avoids interference between tasks, it cannot perform multiple tasks simultaneously. It serves as a reference *upper bound* for model merging approaches.

- **Traditional MTL** aggregates the original training data from all tasks to train a single multi-task model.

**ii) Test-time training-based methods:**

- **AdaMerging** (Yang et al., 2024b) leverages an unlabeled test set to adaptively learn the merging coefficients at either a layer-wise or task-wise level in Task Arithmetic.

- **AdaMerging++** (Yang et al., 2024b) an enhanced version of AdaMerging, integrates the principles of Ties-Merging (Yadav et al., 2023).

- **Surgery** (Yang et al., 2024a) introduces a feature transformation module, trained to align features during the merging process. In this work, we adopt the basic version of Surgery combined with task arithmetic for evaluation

**iii) Training-free methods:**

- **Weight Averaging** directly averages model parameters from multiple tasks into a single model, enabling multi-task learning without additional training.
- **Fisher Merging** (Matena & Raffel, 2022) leverages the Fisher information matrix to assess parameter importance, merging model parameters based on this importance.
- **RegMean** (Jin et al., 2023) refines weight matrices by adjusting and linearly combining rows, utilizing statistical information derived from the training data.
- **Task Arithmetic** (Ilharco et al., 2023b) introduces the concept of a "task vector," defined as the difference between fine-tuned model parameters and pre-trained model parameters. Multiple task vectors are then combined and added to the pre-trained model to facilitate multi-task learning.
- **Ties-Merging** (Yadav et al., 2023) eliminates unimportant parameters from the task vector and resolves sign conflicts among parameters, reducing interference during the final task vector merging process.

**iv) Our methods:**

- **TATR**. This is the core method introduced in our work, which applies task arithmetic within the trust region.
- **TATR zero-shot** A zero-shot variant of TATR that utilizes task vectors to estimate the gradient as described in Eq.(7). Other zero-shot variations follow a similar approach.
- **Ties-Merging & TATR** integrates TATR into the Ties-Merging framework by applying TATR's mask on the task vectors after processing them with Ties-Merging.
- **AdaMerging & TATR** plugged TATR into AdaMerging, where TATR is applied prior to training the AdaMerging coefficients.
- **Surgery & TATR**. Similarly, TATR is integrated into Surgery by applying it before training the additional modules introduced by Surgery.

# B  VISUALIZATION OF LOSS LANDSCAPE

## B.1  METHODOLOGY FOR VISUALIZING THE LOSS LANDSCAPE

In this section, we outline the methodology for visualizing the loss landscape, which involves three key steps:

**Task vector decomposition**. To effectively visualize the loss landscape, we first decompose a task-specific vector into three essential components: a positive component (aligned with the gradient direction), a negative component (opposed to the gradient), and an orthogonal component (orthogonal to the gradient). These components are derived by analyzing the relationship between the task vector and the gradient, as detailed in Section 5. This decomposition allows us to explore how different aspects of the task vector interact with the gradient, each contributing uniquely to the overall optimization behavior.

**Constructing the 2D plane**. We arrange these three components in a 2D coordinate system, with the positive component anchored at (0,1), the negative component at (0,0), and the orthogonal component at (1,0). Additionally, we project the parameters of the pre-trained model onto this plane using linear combinations of the three components. Although this projection is an approximation, as the pre-trained model's parameters may not lie perfectly within the plane defined by these vectors, it provides sufficient insight into the interaction between task vectors and knowledge conflict.

**Contour plot generation**. Finally, we sample points across the plane by selecting coordinates within the range of [-0.2, 1.2] for both axes at intervals of 0.1. For each sampled point, we adjust

the model's parameters accordingly and compute the corresponding loss value. These loss values are then used to generate a contour plot, providing a visual representation of the loss landscape.

## B.2 Loss Landscape for Each Individual Task

In this section, we present the loss landscape for each individual task, along with the components of task vectors from the other seven tasks within the landscape. From Figure 5, we observe the same patterns as described in the manuscript. Specifically, the positive component tends to ascend along the gradient direction, while the negative component, despite aligning with the gradient descent direction, often overshoots local optima, leading to performance degradation. The orthogonal component, in general, shows little sensitivity to performance changes. These findings further support the generality of our conclusions and provide additional evidence for the effectiveness of the TATR method.

## B.3 Loss Landscape for All Tasks

Furthermore, we visualize the overall loss landscape across all tasks, including the components of the task vectors. We present the loss landscape under different mask ratios. As shown in Figure 6, we observe similar patterns: both the positive and negative components negatively impact the model's overall multi-task performance, leading to knowledge conflicts. In contrast, the orthogonal component contributes to improving the model's multi-task capability.

# C Additional Experiments

## C.1 Comparison on ViT-B/16

Table 3 presents the results of various model merging methods using the ViT-B/16 architecture. As we can see, TATR significantly improves the multi-task performance of Task Arithmetic, raising the average performance from 73.8% to 77.0%. Additionally, the zero-shot version also provides a certain degree of improvement, ultimately reaching 74.1%.

Table 3: Multi-task performance when merging ViT-B/16 models on eight tasks.

| Method | SUN397 | Cars | RESISC45 | EuroSAT | SVHN | GTSRB | MNIST | DTD | Avg Acc |
|---|---|---|---|---|---|---|---|---|---|
| Pre-trained | 63.8 | 64.6 | 65.7 | 54.5 | 52.0 | 43.3 | 51.7 | 45.1 | 55.0 |
| Individual | 81.8 | 86.8 | 96.9 | 99.7 | 97.8 | 99.1 | 99.7 | 82.0 | 92.9 |
| Weight Averaging | 67.7 | 70.0 | 75.3 | 79.5 | 74.9 | 60.1 | 94.4 | 43.8 | 70.7 |
| Fisher Merging | 68.5 | 69.9 | 75.2 | 80.4 | 73.2 | 61.2 | 94.5 | 50.7 | 71.7 |
| RegMean | 69.1 | 71.6 | 77.6 | 88.8 | 83.7 | 70.2 | 96.9 | 54.6 | 76.6 |
| Task Arithmetic | 61.1 | 65.9 | 74.0 | 76.2 | 88.0 | 73.9 | 98.4 | 53.0 | 73.8 |
| Ties-Merging | 69.1 | 72.5 | 80.5 | 84.0 | 85.0 | 71.5 | 98.1 | 54.9 | 77.0 |
| **TATR zero-shot (Ours)** | 60.5 | 63.4 | 73.0 | 78.8 | 88.4 | 75.8 | 98.4 | 54.6 | 74.1 |
| **TATR (Ours)** | 67.4 | 70.4 | 77.9 | 81.7 | 87.6 | 77.2 | 98.3 | 55.6 | 77.0 |

## C.2 Generalization Comparison

This section explores the generalization ability of TATR. Specifically, we merge models using task vectors from six tasks and evaluate their performance on two unseen tasks. We conduct two experiments: in the first, MNIST and EuroSAT are set as unseen tasks, while in the second, RESISC45 and SVHN are treated as unseen. The results in Table 4 show that TATR outperforms Task Arithmetic on the unseen datasets, with an average performance improvement of 0.8% and 1.3%, respectively. This improvement in generalization is attributed to TATR's ability to handle knowledge conflicts, ensuring that model updates move toward a more globally optimal direction.

## C.3 Analysis of Exemplar Number

In this section, we further investigate the sensitivity of TATR to the number of exemplars. Table 5 reports the merging performance with varying exemplar numbers. As shown, the zero-shot version consistently outperforms Task Arithmetic across all tasks, achieving an average performance

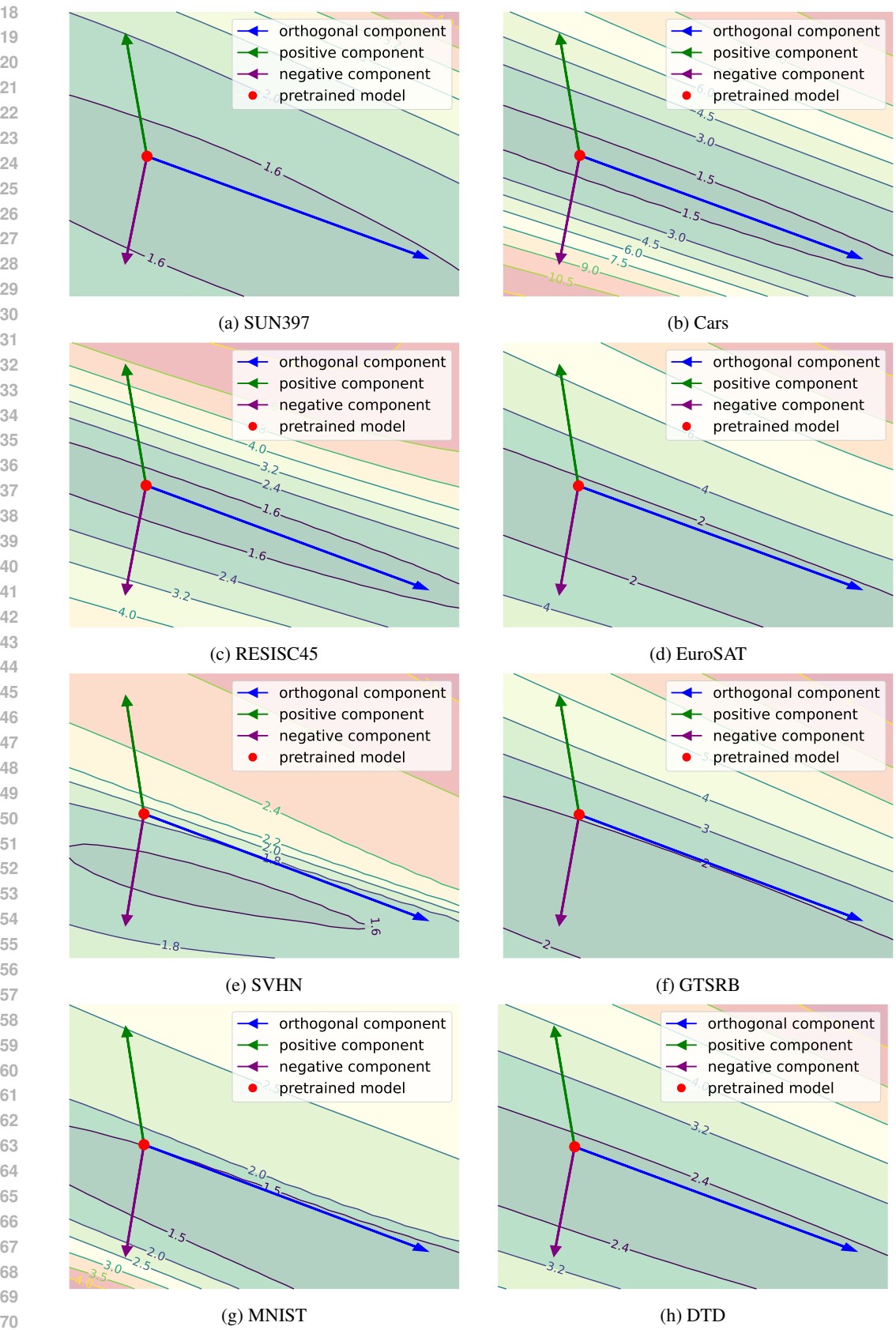

Figure 5: Loss landscape for each dataset and the components of the cumulative task vector from the remaining seven datasets.

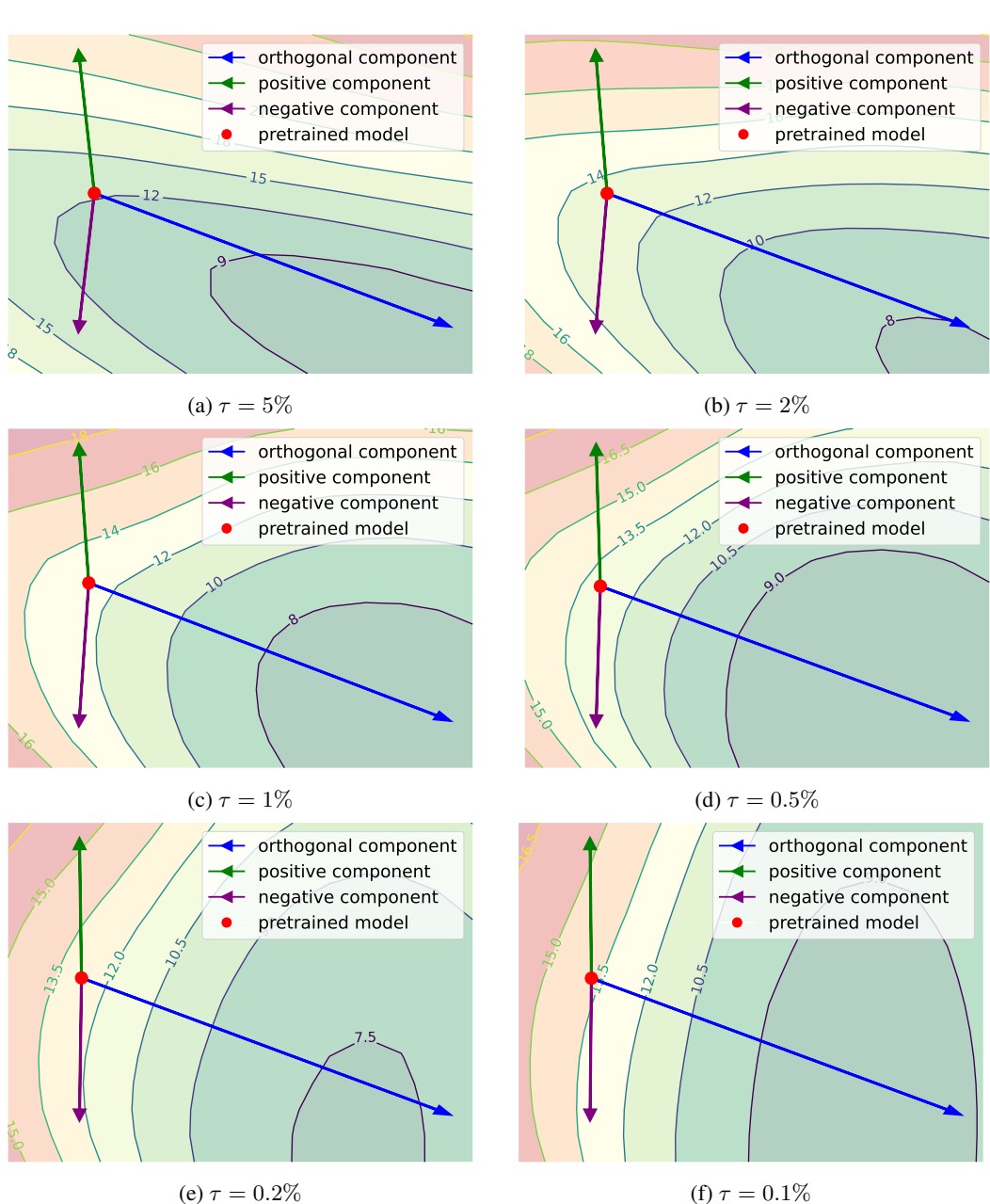

(a) $\tau = 5\%$

(b) $\tau = 2\%$

(c) $\tau = 1\%$

(d) $\tau = 0.5\%$

(e) $\tau = 0.2\%$

(f) $\tau = 0.1\%$

Figure 6: Loss landscape for each datasets and the components of the cumulative task vector from the remaining seven datasets.

Table 4: Generalization results on two unseen tasks when merging ViT-B/16 models on six tasks.

| Method | SUN397 | Cars | RESISC45 | DTD | SVHN | GTSRB | Avg Acc | MNIST | EuroSAT | Avg Acc |
|---|---|---|---|---|---|---|---|---|---|---|
| Task Arithmetic | 63.3 | 62.4 | 75.1 | 57.8 | 84.6 | 80.4 | 70.6 | 77.2 | 46.2 | 61.7 |
| **TATR (Ours)** | 66.0 | 64.1 | 77.9 | 60.1 | 83.9 | 81.8 | 72.3 | 77.2 | 47.7 | 62.5 |

| Method | SUN397 | Cars | GTSRB | EuroSAT | DTD | MNIST | Avg Acc | RESISC45 | SVHN | Avg Acc |
|---|---|---|---|---|---|---|---|---|---|---|
| Task Arithmetic | 64.0 | 64.0 | 75.2 | 87.7 | 57.0 | 95.7 | 73.9 | 52.3 | 44.9 | 51.1 |
| **TATR (Ours)** | 66.5 | 65.2 | 76.8 | 87.9 | 59.5 | 95.6 | 75.3 | 54.7 | 50.0 | 52.4 |

improvement of 1.7%. In the one-shot scenario, TATR significantly boosts performance, with an average increase of 3.4% per task, nearing optimal performance. As the number of exemplars increases, the performance improves across all tasks, obtaining the best performance at the number 16.

Table 5: Impact of the number of exemplars when merging ViT-B/32 models on eight tasks.

| Method | Exemplar number | SUN397 | Cars | RESISC45 | EuroSAT | SVHN | GTSRB | MNIST | DTD | Avg Acc |
|---|---|---|---|---|---|---|---|---|---|---|
| Task Arithmetic | - | 55.2 | 54.9 | 66.7 | 78.9 | 80.2 | 69.7 | 97.3 | 50.4 | 69.1 |
| **TATR zero-shot (Ours)** | - | 59.0 | 56.6 | 69.2 | 80.2 | 79.0 | 70.5 | 97.0 | 53.5 | 70.6 |
| **TATR (Ours)** | 1 | 62.0 | 59.0 | 71.6 | 81.8 | 80.3 | 72.4 | 96.9 | 54.7 | 72.3 |
| **TATR (Ours)** | 2 | 62.3 | 59.2 | 71.6 | 81.5 | 80.5 | 72.4 | 97.0 | 55.4 | 72.5 |
| **TATR (Ours)** | 4 | 62.3 | 59.3 | 71.8 | 82.4 | 80.5 | 72.7 | 97.0 | 55.1 | 72.6 |
| **TATR (Ours)** | 8 | 62.6 | 59.3 | 72.2 | 82.3 | 80.1 | 72.6 | 97.0 | 55.3 | 72.7 |
| **TATR (Ours)** | 16 | 62.7 | 59.5 | 72.3 | 82.4 | 80.4 | 72.6 | 97.0 | 55.3 | 72.8 |
| **TATR (Ours)** | 32 | 62.7 | 59.5 | 72.4 | 82.4 | 80.4 | 72.5 | 97.0 | 55.4 | 72.8 |
| **TATR (Ours)** | 64 | 62.7 | 59.3 | 72.3 | 82.5 | 80.4 | 72.7 | 97.0 | 55.4 | 72.8 |
| **TATR (Ours)** | 128 | 62.7 | 59.4 | 72.3 | 82.5 | 80.4 | 72.7 | 97.0 | 55.4 | 72.8 |

Table 6: Comparison with different sensitivities of TATR when merging ViT-B/32 models on eight tasks.

| Method | Sensitivity | SUN397 | Cars | RESISC45 | EuroSAT | SVHN | GTSRB | MNIST | DTD | Avg Acc |
|---|---|---|---|---|---|---|---|---|---|---|
| Pre-trained | - | 63.8 | 64.6 | 65.7 | 54.5 | 52.0 | 43.3 | 51.7 | 45.1 | 55.0 |
| TATR positive | $\frac{1}{K(K-1)} \sum_{i \neq j} \nabla_\theta L_j \left(\theta_{\text{pre}}\right) \odot \Delta_i$ | 60.0 | 54.3 | 44.8 | 9.3 | 18.9 | 14.3 | 17.1 | 39.4 | 32.3 |
| TATR negative | $-\frac{1}{K(K-1)} \sum_{i \neq j} \nabla_\theta L_j \left(\theta_{\text{pre}}\right) \odot \Delta_i$ | 29.5 | 11.5 | 22.6 | 30.0 | 65.5 | 40.0 | 83.3 | 30.5 | 39.1 |
| TATR ntk | $\frac{1}{K(K-1)} \sum_{i \neq j} \left|\nabla_\theta L_j \left(\theta_{\text{pre}}\right)\right| \odot \left|\nabla_\theta L_i \left(\theta_{\text{pre}}\right)\right|$ | 61.8 | 59.0 | 71.5 | 81.3 | 81.3 | 72.9 | 97.2 | 55.3 | 72.5 |
| TATR zero-shot | $\frac{1}{K(K-1)} \sum_{i \neq j} \left|\Delta_j\right| \odot \left|\Delta_i\right|$ | 59.0 | 56.6 | 69.2 | 80.2 | 79.0 | 70.5 | 97.0 | 53.5 | 70.6 |
| TATR | $\frac{1}{K(K-1)} \sum_{i \neq j} \left|\nabla_\theta L_j \left(\theta_{\text{pre}}\right)\right| \odot \left|\Delta_i\right|$ | 62.7 | 59.3 | 72.3 | 82.3 | 80.5 | 72.6 | 97.0 | 55.4 | 72.8 |

## C.4 ANALYSIS OF SENSITIVITY FOR KNOWLEDGE CONFLICT

In this section, we explore various forms of conflict sensitivity in knowledge sharing. Specifically, we examine the following five approaches:

- **TATR positive**: This is calculated as the product between the task vector and the gradient. It promotes the merging of the components in the task vector that align with the gradient's ascent direction. The sensitivity of TATR positive is formulated as:

$$\frac{1}{K(K-1)} \sum_{i \neq j} \nabla_\theta L_j \left(\theta_{\text{pre}}\right) \odot \Delta_i.$$

- **TATR negative**: This is computed as the product between the negative task vector and the gradient. It encourages the merging of the components in the task vector that follow the gradient's descent direction. The sensitivity of TATR negative is formulated as:

$$-\frac{1}{K(K-1)} \sum_{i \neq j} \nabla_\theta L_j \left(\theta_{\text{pre}}\right) \odot \Delta_i.$$

- **TATR ntk**: This approach computes the product of the absolute values of gradients from different tasks, analogous to the Neural Tangent Kernel (NTK) (Jacot et al., 2018). It

measures the influence of model updates for one task on another. Specifically, TATR ntk utilizes the following sensitivity for knowledge conflict:

$$\frac{1}{K(K-1)} \sum_{i \neq j} |\nabla_\theta L_j (\theta_{\text{pre}})| \odot |\nabla_\theta L_i (\theta_{\text{pre}})| .$$

- **TATR Zero-shot**: The zero-shot variant calculates the product of the absolute values between task vectors from different tasks:

$$\frac{1}{K(K-1)} \sum_{i \neq j} |\Delta_j| \odot |\Delta_i| .$$

- **TATR (Standard Version)**: The standard version computes the product of the absolute values of the task vector and the gradient. It encourages the fusion of the components in the task vector that are orthogonal to the gradient. The sensitivity of TATR is calculated as follows:

$$\frac{1}{K(K-1)} \sum_{i \neq j} |\nabla_\theta L_j (\theta_{\text{pre}})| \odot |\Delta_i| .$$

Table 6 reports the multi-task performance of these methods when merging ViT-B/32 models across eight tasks. It is evident that both TATR Negative and TATR Positive result in a significant performance drop, indicating severe knowledge conflicts. In contrast, the standard TATR method effectively improves the performance of the pre-trained model across tasks, significantly mitigating knowledge conflicts. While the zero-shot and NTK variants exhibit slight performance degradation compared to TATR, they also demonstrate the ability to alleviate knowledge conflicts, suggesting that task vectors can approximate gradients to some extent.

