# OpenReview forum: "Task Arithmetic in Trust Region: A Training-Free Model Merging Approach to Navigate Knowledge Conflicts"
_ICLR.cc/2025/Conference — Submitted to ICLR 2025_

### Official Review · Reviewer_V85r · 2024-10-29

**Soundness:** 2
**Presentation:** 3
**Contribution:** 2
**Rating:** 6
**Confidence:** 4

**Summary:**

This paper introduces Task Arithmetic in Trust Region (TATR), a training-free approach for multi-task model merging. By defining a trust region, TATR restricts merging to parameters with minimal influence on task-specific losses, aiming to reduce knowledge conflicts. The method identifies that gradient-aligned task vector components are primary sources of conflict and manages the merging process within a controlled parameter subset. TATR is designed to integrate easily with existing merging methods like AdaMerging and Ties-Merging without additional training, providing an efficient approach to address knowledge conflicts in model merging.

**Strengths:**

- This paper proposes a method, called Task Arithmetic in Trust Region (TATR), that solves the issue of knowledge conflicts. It explores this issue from the dimensions in the model parameter space that cause only small changes in the task-specific losses.
- The paper with clear explanations about the motivation behind Task Arithmetic in Trust Region (TATR).
- The experiments covering multiple datasets demonstrate that TATR improves multi-task performance.

**Weaknesses:**

- Regarding knowledge conflict, which has been previously explored in works such as [1] Cross-task Knowledge Distillation in Multi-Task Recommendation, [2] Multi-Task Distillation: Towards Mitigating Negative Transfer in Multi-Task Learning, [3] Conflict-Averse Gradient Descent for Multi-Task Learning, [4] Hacking Task Confounder in Meta-Learning, and [5] Parameter Competition Balancing for Model Merging, with the similar method (model merging from the parameter perspective). How does this work differ from these studies? Are there any new insights provided? **This is my biggest concern.**

- The paper employs an orthogonal component for model merging, intuitively, this approach might introduce task-irrelevant noise into the model, and I find it challenging to understand how such noise could enhance model performance theoretically.

- The proposed method performs poorly on the Cars dataset. Could the authors provide an analysis of why this might be the case?

- (Minor) In Table 1, the authors should highlight the best and second-best performance values, as their absence significantly affects readability. Additionally, indicating the number of datasets on which the proposed method achieved the best performance would facilitate a more straightforward quantitative assessment of its advantages.

**Questions:**

Please see **Weaknesses**.

---

> ### Author Response · Authors · 2024-11-21
> **Response to V85r Part 1**
>
> **Q: What are the differences and new insights of knowledge conflict between existing research [1-5]?**
>
> A: Thank you for the important question. Works such as [1-4] address negative transfer focusing on the **dynamic** gradient interference during multi-task training. Our work addresses model merging from **static** task vectors. For example, the large step sizes in task vectors introduce a unique set of problems not present during multi-task training. We have added the citation of these papers in the discussion of *the second paragraph of the Introduction* of our manuscript.
>
> **[5] is contemporaneous with our work.** It will be published at NeurIPS 2024 on 10 Dec 2024, which is after the ICLR submission deadline (01 Oct 2024). In addition, [5] is substantially different with our work in terms of methodology and objectives. [5] eliminates components that have **small** products between two task vectors but TATR masks components with **big** absolute products between task vectors and gradients. In terms of objectives, [5] tends to make task vectors **aligned**, whereas we aim to make them **orthogonal**. Beyond the difference in objectives, our approach is a **lightweight, plug-and-play module**, distinguishing it from the method in [5], which contains multiple techniques for test-time adaptation.
>
> We hope this response clarifies the unique contributions of this work, and will incorporate these comparisons in the revised version. Thank you again for bringing this to our attention.
>
> [1] Yang, Chenxiao, et al. "Cross-task knowledge distillation in multi-task recommendation." *Proceedings of the AAAI conference on artificial intelligence*. Vol. 36. No. 4. 2022.
>
> [2] Meng, Ze, Xin Yao, and Lifeng Sun. "Multi-task distillation: Towards mitigating the negative transfer in multi-task learning." *IEEE International Conference on Image Processing (ICIP)*. 2021.
>
> [3] Liu, Bo, et al. "Conflict-averse gradient descent for multi-task learning." *Advances in Neural Information Processing Systems.* 2021.
>
> [4] Wang J, Ren Y, Song Z, et al. "Hacking task confounder in meta-learning". *IJCAI*. 2024.
>
> [5] Guodong, D. U., et al. "Parameter Competition Balancing for Model Merging." *Advances in Neural Information Processing Systems*. 2024.

---

> ### Author Response · Authors · 2024-11-21
> **Response to V85r Part 2**
>
> **Q: Orthogonal components may introduce task-irrelevant noise and harm to model performance.**
>
> A: Thank you for your insightful question. Intuitively, parameter changes in directions orthogonal to the gradient directions are expected to introduce less task-irrelevant noise than changes in the directions parallel to the gradient.
>
> We empirically validate if TATR reduces interference between tasks by comparing TATR with TA when a new task vector is introduced to the mix of task vectors. Specifically, we quantify the test accuracy changes for a given task between incorporating and excluding a task vector. Each table entry in the following (corresponding to column task $j$ and row task $i$)  displays the difference in task $j$ accuracy when including all task vectors versus excluding the vector of the task $i$., i.e., $ACC_j( \theta_{MTL}(\Delta_1, \dots, \Delta_K) ) - ACC_j( \theta_{MTL}(\Delta_1, \dots, \Delta_{i-1}, \Delta_{i+1}, \dots, \Delta_K))$.
>
> TATR has a notably smaller interference between tasks compared to TA. This confirms that TATR introduces limited task-irrelevant noise.
>
> | Method |          | SUN397    | Cars      | RESISC45  | EuroSAT   | SVHN      | GTSRB     | MNIST     | DTD       |
> | ------ | -------- | --------- | --------- | --------- | --------- | --------- | --------- | --------- | --------- |
> | TA     | SUN397   | -         | -3.15     | -2.65     | -1.89     | -1.20     | -2.75     | -0.03     | -2.18     |
> | TATR   | SUN397   | -         | **-2.45** | **-1.73** | **-0.70** | **-0.45** | **-1.03** | **0.02**  | **-0.69** |
> | TA     | Cars     | -4.41     | -         | -3.48     | -1.78     | -2.37     | -2.48     | **-0.15** | -1.86     |
> | TATR   | Cars     | **-1.69** | -         | **-1.87** | **-0.52** | **-1.54** | **-1.69** | -0.17     | **-1.06** |
> | TA     | RESISC45 | -4.30     | -3.38     | -         | 3.48      | -3.42     | -4.89     | **-0.32** | -3.40     |
> | TATR   | RESISC45 | **-2.00** | **-2.39** | -         | **4.85**  | **-2.27** | **-3.16** | -0.33     | **-2.07** |
> | TA     | EuroSAT  | -2.47     | -2.08     | **0.10**  | -         | -5.55     | -3.66     | -0.73     | **-2.45** |
> | TATR   | EuroSAT  | **-1.58** | **-1.89** | -1.35     | -         | **-3.93** | **-2.50** | **-0.47** | -2.61     |
> | TA     | SVHN     | -5.20     | -5.66     | -6.37     | -9.85     | -         | -0.21     | 2.62      | -3.09     |
> | TATR   | SVHN     | **-2.32** | **-3.94** | **-3.92** | **-7.41** | -         | **1.08**  | **2.65**  | **-2.55** |
> | TA     | GTSRB    | -7.22     | -4.96     | -7.27     | -9.15     | 3.49      | -         | -0.56     | -4.84     |
> | TATR   | GTSRB    | **-3.05** | **-3.02** | **-4.17** | **-5.89** | **4.56**  | -         | **-0.52** | **-3.35** |
> | TA     | MNIST    | -6.23     | -5.07     | -7.25     | -9.22     | 1.78      | -6.26     | -         | -5.0      |
> | TATR   | MNIST    | **-2.70** | **-3.21** | **-4.79** | **-6.30** | **2.38**  | **-4.13** | -         | **-3.51** |
> | TA     | DTD      | -5.09     | -3.48     | -3.25     | -3.33     | -1.20     | -2.03     | -0.13     | -         |
> | TATR   | DTD      | **-1.98** | **-2.14** | **-1.83** | **-1.96** | **-0.16** | **-0.55** | **0.01**  | -         |
>
> **Q: Why does TATR perform poorly on the Cars dataset?**
>
> A: We notice that the Cars dataset is challenging for **all** TA-based methods. For instance, under the ViT-B/32 architecture, even AdaMerging++, a test-time training-based approach, struggles to outperform simpler non-TA-based methods like Fisher Merging.
>
> A possible explanation is that the Cars dataset has larger knowledge conflicts than other datasets. As highlighted in *Figure 4* of the manuscript, the knowledge conflict of Cars is markedly higher than that of other datasets. This heightened knowledge conflict guides TATR to mask a substantial knowledge of the Cars task vector, inadvertently discarding critical task-specific information. As a result, the merged model struggles to perform well on the Cars dataset.
>
> **Q: The authors should highlight the best and second-best performance values, and indicate the number of datasets on which the proposed method achieved the best.**
>
> A: Thanks for your suggestion. We have uploaded a revised manuscript where these changes are made.

---

> ### Comment · Reviewer_V85r · 2024-11-25
> **Official Comment by Reviewer V85r**
>
> Thank you for your response, which has addressed most of my concerns, especially regarding the differences about the knowledge transfer concept and methods with prior works (W1 for motivation and concept overlap). Please ensure that the additional experimental results and explanations are incorporated into the work. I am happy to raise my score accordingly.

---

> > ### Author Response · Authors · 2024-11-25
> >
> > Thank you for your feedback. We deeply appreciate your reconsideration of the rating.
> >
> > We will ensure that the additional experiments and discussions are incorporated into the final version.

---

### Official Review · Reviewer_H7en · 2024-10-30

**Soundness:** 2
**Presentation:** 3
**Contribution:** 2
**Rating:** 6
**Confidence:** 3

**Summary:**

This paper identifies the phenomenon of task conflict in multi-task model merging, primarily occurring in certain components of task vectors that are aligned with the direction of the task gradients. Based on this finding, this paper defines a trust region (i.e., the task vector components orthogonal to the gradient) and introduces the TATR method, which performs parameter merging only within this trust region to alleviate knowledge conflicts during model merging. As a plug-and-play approach, TATR can be combined with previous methods, and extensive evaluation on diverse datasets demonstrates its effectiveness.

**Strengths:**

1. The paper is well-written and easily understandable.

2. TATR demonstrates promising performance on ViT-B/32.

3. The method proposed in this paper is interesting and can be combined with existing methods, which enhance the practical value of this method.

**Weaknesses:**

1. The motivation does not seem to align with the proposed method. The method's motivation is to select components of task vectors orthogonal to other task gradients for model merging, but the approach instead selects components with smaller values (or norms) for merging.

2. The paper introduces a zero-shot version of TATR. Although Line 313 claims, "Although there may be estimation errors, this approach still offers performance improvements for TA-based methods ..." I observed in the experiments that this version of TATR in Table 1 for test-time training based methods does not yield performance gains. Therefore, I believe this sentence should be revised.

3. TATR uses an exemplar set to calculate the gradient, and I'm interested in whether the samples here are randomly sampled or selected in some way. If it is random sampling, can you analyze if the different samples have an effect on the selection of the trust region?

4. Despite promising results on the ViT-B/32 model, the performance of TATR is limited on the ViT-L/14.

5. Line 414, "Table 5 and Table 2" is wrong.

**Questions:**

1. Robustness analysis for different samples.

2. What is the difference between TATR and the direct removing components with large high magnitude of the task vector?

3. Can you analyze why there is a difference in performance on ViT-B/32 and ViT-L/4?

---

> ### Author Response · Authors · 2024-11-21
> **Response to H7en Part 1**
>
> **Q: Does TATR directly remove components with a high magnitude of the task vector? Is TATR aligned with the motivation that selecting components of task vectors orthogonal to other task gradients?**
>
> A: TATR does not simply “select components with smaller values (or norms) for merging.” Instead, we select components based on **smaller absolute products** between the task vector and the gradient (*Definition 2)*. Selecting vector components with small absolute products with the gradient vector is a sufficient condition for the update vector to be orthogonal to the gradient vector. Therefore, selecting components with smaller absolute products aligns the motivation of selecting components of task vectors orthogonal to other task gradients
>
> To demonstrate that our smaller absolute products do not rely on selecting components with smaller values, we conducted a straightforward experiment: we removed high-magnitude components from task vectors based on TA in a removal ratio τ, denoted as RM(τ). The table below shows that the new method is substantially worse than TATR.
>
> | Method   | TA   | RM(0.1%) | RM(0.2%) | RM(0.5%) | RM(1%) | RM(2%) | RM(5%) | RM(10%) | TATR |
> | -------- | ---- | -------- | -------- | -------- | ------ | ------ | ------ | ------- | ---- |
> | ViT-B/32 | 69.1 | 69.4     | 69.4     | 69.3     | 69.3   | 69.2   | 68.7   | 68.0    | 72.8 |
> | ViT-L/14 | 84.5 | 84.3     | 84.2     | 84.1     | 83.9   | 83.7   | 83.1   | 82.3    | 85.3 |
>
> **Q: The claim about the zero-shot version of TATR is not accurate since TATR does not yield performance gains in all cases.**
>
> A: Thanks for your suggestion. Considering that the zero-shot version of TATR shows improvements in 7 out of 9 scenarios (including ViT-B/16), we revise the statement to "Although there may be estimation errors, this approach still offers performance improvements for TA-based methods in **most** scenarios."

---

> ### Author Response · Authors · 2024-11-21
> **Response to H7en Part 2**
>
> **Q: Robustness analysis for different sample sets.**
>
> A: To evaluate the robustness of exemplar selection, we repeated the TATR experiment five times using randomly selected exemplar sets. The table below presents the mean and standard deviation over five runs with varying numbers of examples, evaluated on both the ViT-B/32 and ViT-L/14 architectures. As can be seen, TATR demonstrates consistently robust performance. Notably, the model achieves both strong performance and robustness with just 16 samples.
>
> | Number of exemplars | 1              | 2              | 4              | 8              | 16             | 32             |
> | ------------------- | -------------- | -------------- | -------------- | -------------- | -------------- | -------------- |
> |                     |                |                | **ViT-B/32**   |                |                |                |
> | Random batch 1      | 72.3           | 72.6           | 72.7           | 72.8           | 72.8           | 72.8           |
> | Random batch 2      | 72.6           | 72.7           | 72.8           | 72.7           | 72.8           | 72.8           |
> | Random batch 3      | 72.2           | 72.5           | 72.7           | 72.7           | 72.8           | 72.8           |
> | Random batch 4      | 72.2           | 72.5           | 72.5           | 72.8           | 72.8           | 72.8           |
> | Random batch 5      | 72.2           | 72.8           | 72.6           | 72.8           | 72.6           | 72.7           |
> | Average             | 72.30$\pm$0.15 | 72.62$\pm$0.12 | 72.66$\pm$0.10 | 72.76$\pm$0.05 | 72.76$\pm$0.08 | 72.78$\pm$0.04 |
> |                     |                |                | **ViT-L/14**   |                |                |                |
> | Random batch 1      | 85.0           | 85.2           | 85.3           | 85.2           | 85.2           | 85.3           |
> | Random batch 2      | 85.0           | 85.1           | 85.2           | 85.2           | 85.2           | 85.4           |
> | Random batch 3      | 84.9           | 85.1           | 85.1           | 85.3           | 85.3           | 85.3           |
> | Random batch 4      | 85.2           | 85.1           | 85.2           | 85.2           | 85.2           | 85.3           |
> | Random batch 5      | 84.8           | 85.2           | 85.3           | 85.2           | 85.3           | 85.3           |
> | Average             | 84.98$\pm$0.13 | 85.14$\pm$0.05 | 85.22$\pm$0.07 | 85.22$\pm$0.04 | 85.24$\pm$0.05 | 85.32$\pm$0.04 |
>
> **Q: Why is there a difference in performance between ViT-B/32 and ViT-L/4?**
>
> A: The differences in performance between ViT-B/32 and ViT-L/14 can be attributed to the higher capacity of ViT-L/14. ViT-L/14 features a larger number of parameters and a smaller patch size (14x14 vs. 32x32), enabling finer-grained feature extraction and richer task-specific representations. These properties mean that different task vectors for ViT-L/14 may not conflict as easily.
>
> This inherent strength of ViT-L/14 limits the room for improvement from model merging methods. For instance, methods like Ties-Merging see a gain of 3.3% on ViT-B/32 (from 69.1% to 72.4%) compared with TA, but only 1.5% on ViT-L/14 (from 84.5% to 86.0%). Similarly, Surgery achieves 11.8% improvement on ViT-B/32 (from 69.1% to 80.9%) but only 4.5% on ViT-L/14 (from 84.5% to 89.0%).
>
> The higher capacity of ViT-L/14  also mitigates the knowledge conflicts, which reduces the relative effectiveness of TATR.
>
> **Q: In line 414, "Table 5 and Table 2" should be "Table 1 and Table 2".**
>
> A: Thanks for your suggestion. We have uploaded a revised manuscript where this typo has been corrected.

---

> > ### Comment · Reviewer_H7en · 2024-11-21
> >
> > Thank you very much for your reply, which solved most of my questions.
> >
> > But I still have two doubts that need your answer:
> >
> > First, although in Definition 2 you stated that the task vector with the smaller absolute product of the gradient is selected, in Equation 8 it is directly estimated as the product of two absolute vectors, which seems to mean that the gradient and task vector with smaller norm are selected. Can you further analyze the rationality of this approximation method? At the same time, I am also curious about the experimental effect of directly selecting smaller components.
> >
> > Second, I am very interested in the robustness of this algorithm to the case of unbalanced sample distribution.
> >
> > Thank you again for your reply. If you can answer these two questions, I will be very happy to modify the score.

---

> > > ### Author Response · Authors · 2024-11-22
> > > **Response to the Reply of H7en**
> > >
> > > We appreciate your feedback. The answers to your questions are as follows.
> > >
> > > **Q: Does Equation 8 mean that components with both the gradient and task vector with smaller norms are selected? Is Equation 8 approximated to *Definition 2* of the manuscript?**
> > >
> > > A: Equation 8 is *identical* to Definition 2 because $\forall a,b \in \mathbb{R}, \ |ab|=|a||b|$. No approximation is necessary.
> > >
> > > **Q: How about directly selecting smaller components before merging?**
> > >
> > > A: In the earlier *Response to H7en Part 1*, we present the results of RM(τ) by removing components with larger values in the first table.
> > >
> > > Based on your feedback, we add experiments with a high removal ratio τ for RM(τ), which selects only a small number of components for model merging. The table below shows that selecting fewer components has no benefit for TA.
> > >
> > > | Method | TA | RM(50%) | RM(80%) | RM(90%) | RM(95%) | RM(98%) | RM(99%) |  $\theta_{pre}$ |
> > > | --- | --- | --- | --- | --- | --- | --- | --- | --- |
> > > | ViT-B/32 | 69.1 | 66.3 | 65.4 | 63.3 | 58.6 | 52.9 | 44.3 | 48.0 |
> > > | ViT-L/14 | 84.5 | 80.8 | 79.9 | 77.2 | 73.8 | 69.9 | 67.8 | 64.5 |
> > >
> > > **Q: robustness of TATR with an unbalanced exemplar set.**
> > >
> > > A: The first table in *Response to H7en Part 2* has shown our results with **only one exemplar**, which is already a highly unbalanced scenario, as the exemplar set contains data solely from a single class.
> > >
> > > We further conduct experiments by selecting 16 samples from **only one class** in each dataset to construct the exemplar set. The following table highlights the robustness of our method in unbalanced scenarios.
> > >
> > > | exemplar set | ViT-B/32 | ViT-B/32 | ViT-L/14 | ViT-L/14 |
> > > | --- | --- | --- | --- | --- |
> > > | distribution of exemplar | unbalanced | balanced | unbalanced | balanced |
> > > | Random batch 1 | 72.9 | 72.8 | 85.1 | 85.2 |
> > > | Random batch 2 | 72.7 | 72.8 | 85.1 | 85.2 |
> > > | Random batch 3 | 72.7 | 72.8 | 85.3 | 85.3 |
> > > | Random batch 4 | 72.8 | 72.8 | 85.2 | 85.2 |
> > > | Random batch 5 | 72.8 | 72.6 | 85.1 | 85.3 |
> > > | Average | 72.78$\pm$0.07 | 72.76$\pm$0.08 | 85.16$\pm$0.08 | 85.24$\pm$0.05 |
> > >
> > > The result may seem unexpected and we offer a hypothetical explanation. Speculatively, gradient vectors of any class may be constrained to a subspace, which is much smaller than the entire parameter space. Using samples from an unbalanced exemplar set still reveals part of this subspace. Since the parameter space is much larger than the said subspace, any random direction orthogonal to any vector in the subspace is likely also orthogonal to the subspace. Hence, any exemplar set can be beneficial to the avoidance of knowledge conflict.

---

> ### Comment · Reviewer_H7en · 2024-11-22
>
> Thank you very much for your reply, which solved most of my questions. The results look very good.
>
> But I still have one more doubt that needs your answer:
>
> In lines 259-265 of the article, you use the element-wise product as the inner product, which is confusing as it seems to conflict with the definition of the inner product, where the output is required to be a scalar. This also makes the definition of 'orthogonal' prone to misunderstanding. Furthermore, I am concerned about whether such a definition is appropriate. So I think maybe there is a more appropriate name to describe it.
>
> Thank you again for your reply. I am very happy to increase the score.

---

> > ### Author Response · Authors · 2024-11-22
> > **Response to the Reply of H7en**
> >
> > Thank you for your valuable feedback and positive score.
> >
> > **Q: Terms defined in lines 259-265 such as “inner product” and “orthogonal components” may be unsuitable.**
> >
> > We agree that the use of "inner product" should be clarified, and we have revised the manuscript accordingly.
> >
> > Regarding the term **"Orthogonal component"**, we prefer to keep it for two main reasons:
> >
> > 1. $\Delta^{\perp}$is orthogonal to the gradient, since their element-wise product is near zero.
> > 2. We aim to convey the idea of reducing knowledge conflict by incorporating orthogonal task vectors, which helps to align with our concept.
> >
> > We hope this explanation clarifies our choice of terminology. If you have any other concerns about our work, we would greatly appreciate receiving any further comments or suggestions.

---

### Official Review · Reviewer_YLeq · 2024-11-04

**Soundness:** 3
**Presentation:** 3
**Contribution:** 3
**Rating:** 6
**Confidence:** 3

**Summary:**

The paper presents a novel approach to task arithmetic aimed at addressing challenges in multi-task learning. It defines conflicts between tasks as "knowledge conflicts" and introduces a trust region to help mitigate these conflicts.

**Strengths:**

-The paper clearly presents the problem and the proposed method.
- It offers an extensive set of experiments across multiple datasets.
- The paper includes a comprehensive comparison with existing methods to demonstrate the performance of the proposed approach.

**Weaknesses:**

The experiments convincingly demonstrate the effectiveness of the proposed data augmentation method. However, the performance improvements are minimal in most cases.

**Questions:**

See the weakness.

---

> ### Author Response · Authors · 2024-11-21
> **Response to YLeq**
>
> **Q: The experiments convincingly demonstrate the effectiveness of the proposed data augmentation method. However, the performance improvements are minimal in most cases.**
>
> A: Thank you for your feedback and support for our work.
>
> Our method demonstrates good performance gains in training-free cases. For instance, on the ViT-B/32 model, our method provides an average improvement of 3.7% for TA and 0.9% for Tiesmerging. On ViT-L/14, the improvements are 0.8% and 0.5%, respectively. Similarly, on ViT-B/16, our method achieves a notable 3.2% improvement for TA.
>
> AdaMerging and Surgery show high performance gains *over TA*. However, our method is better in the following ways.
>
> 1. TATR can be applied on top of AdaMerging and Surgery and improve these methods further.
> 2. AdaMerging and Surgery require expensive training with substantial data, whereas TATR is a lightweight, plug-and-play module that is completely training-free.

---

### Official Review · Reviewer_vK8u · 2024-11-04

**Soundness:** 2
**Presentation:** 3
**Contribution:** 3
**Rating:** 5
**Confidence:** 4

**Summary:**

This paper introduces the concept of "knowledge conflicts," a static property between task-specific fine-tuned model parameters and base model parameters, identifying it as a key issue in Task Arithmetic. Through analysis, the authors propose a novel trust-region criterion for model merging, called Task Arithmetic in Trust Region (TATR), designed to effectively mitigate these knowledge conflicts. The proposed method can also be applied orthogonally to other Task Arithmetic-based approaches.

**Strengths:**

1. The paper is easy to follow, with "knowledge conflicts" serving as a clear and intuitive motivation. Researchers familiar with "gradient conflicts" between tasks will easily understand it, as the authors clearly differentiate these two concepts.

2. The proposed methods can enhance multi-task performance when used alongside existing Task Arithmetic (TA) methods, demonstrating the practical value of the approach.

**Weaknesses:**

1. The author argues that merging both gradient descent and ascent components would lead to negative performance across multiple tasks, as each component contributes differently—gradient descent causes knowledge conflicts, while gradient ascent leads to overshooting. This claim appears controversial, suggesting that other important criteria or metrics might also play a role in causing knowledge conflicts. If the performance degradation from merging the negative gradient descent is purely due to the large magnitude of the task-specific vectors, could reducing their magnitude make them usable?

2. The authors propose a practical approach by measuring the "Trust Region" through the inner product of task vectors and the gradient of loss, considering each task vector component's effect on merging performance. A question arises: according to Eq. (5), it appears that the inner product of each task pair is summed to calculate the trust region. However, intuitively, it seems that the trust region should vary for each task pair depending on the inherent relations between tasks. The proposed method, however, does not seem to account for this variability across task pairs, instead summing their magnitudes uniformly to determine the trust region.

3. The proposed method is reminiscent of PCGrad [1], which is used in multi-task optimization and projects task vectors to project task vectors into orthogonal space to better align them. Since achieving perfectly orthogonal vectors is impossible when handling multiple tasks, PCGrad randomly mixes gradients to assign priority. This raises a related question in the context of Question 2: When dealing with multiple tasks with distinct objectives, is it always possible to find orthogonal components for task vectors across various tasks? It’s unclear if this can always be guaranteed, and a theoretical analysis may be necessary to further validate the proposed methods and address these potential limitations.

[1] Yu, Tianhe, et al. "Gradient Surgery for Multi-task Learning." Advances in Neural Information Processing Systems 33 (2020): 5824-5836.


I am willing to raise my score if all the weaknesses can be addressed adaptively.

**Questions:**

1. The authors provide informative figures (Figures 2, 5, and 6) to illustrate how updates of orthogonal components can reduce knowledge conflict. However, this effect appears dependent on the homogeneity of the task set, as most experiments are conducted on simple, domain-specific classification tasks, which are relatively homogeneous. This raises questions about the broader validity of Task Arithmetic in more diverse settings. Although most baselines, such as AdaMerging, also use these simpler setups, which explains the authors' choice to some extent, the proposed methods could be tested on benchmarks with heterogeneous task settings—such as Taskonomy, PASCAL-Context, or NYUD—that feature a wider variety of task relations and are commonly used in MTL research.

---

> ### Author Response · Authors · 2024-11-21
> **Response to vK8u Part 1**
>
> **Q: If a large task vector along the gradient descent direction causes overshooting, how about scaling down the magnitude of the task vector?**
>
> A: Thank you for the suggestion. We agree that other important criteria, such as vector magnitudes, could play a critical role in knowledge conflicts. However, finding an appropriate scaling factor for various negative components remains a challenging task. We conducted a straightforward experiment: if a task vector component has the same sign with the corresponding component of the gradient descent vector, we follow your advice to scale it down using a multiplier $\tau$. Otherwise, we mask that component of the task vector.
>
> $m[n] = \tau \ \  \text{if } \sum_{i \neq j} \nabla_\theta L_j( \theta_{\textnormal{pre}})[n] \cdot \Delta_i [n] < 0 \ \  \text{else} \ \  0 $
>
> $\theta_{MTL} = \theta_{pre} + \lambda m \odot \sum_i \Delta_i$
>
> We tried different multipliers and the results are in the following table. The new method is substantially worse than TATR.
>
> |          | TATR | $\tau=0.001$ | $\tau=0.01$ | $\tau=0.1$ | $\tau=0.2$ | $\tau=0.3$ | $\tau=0.5$ | $\tau=0.8$ | $\tau=0.9$ |
> | -------- | ---- | ------------ | ----------- | ---------- | ---------- | ---------- | ---------- | ---------- | ---------- |
> | ViT-B/32 | 72.8 | 45.5         | 43.0        | 40.8       | 37.0       | 43.6       | 38.1       | 45.7       | 42.6       |
> | ViT-L/14 | 85.3 | 64.5         | 67.0        | 68.4       | 68.0       | 62.2       | 69.3       | 67.4       | 65.1       |
>
> **Q: The paper computes the trust region by summing over all tasks. Why not compute a trust region for each pair of tasks?**
>
> A: Our algorithm aims to create a network that can handle *all* tasks of a multi-task problem. Hence, we must update $\theta_{pre}$ with all task vectors and use a trust region from all task vectors. We do not build trust regions for each task pair, because we do not create separate networks for each task pair.
>
> To further illustrate the benefits of the TATR trust region, we do the following experiment. We first compute a TATR-style trust region for each possible pair of tasks, which selects some components of the task vectors. After that, we compute the intersection of the selected components.
>
> Formally, the TATR-style trust region for every task pair $(i, j)$ is defined as $TR_{j|i}$, which contains selecting components of the task vectors satisfying $\forall n\in TR_{j|i} ,\  \{   \big| \nabla_\theta L_j( \theta_{pre})[n] \cdot \Delta_i [n]  \big|  < \epsilon \} $, and then we merge the network weights as $\theta_{pre} + \lambda \sum_k \Delta_k \odot 1_{\{n \in \mathcal{ \hat{TR} }\}}$ where $\hat{TR} =\bigcap_{i, j, i \neq j} TR_{j|i}$.
>
> The following table shows that the new method degrades performance, proving the effectiveness of maintaining a single trust region.
>
> | Backbone | Method                                  | SUN397 | Cars | RESISC45 | EuroSAT | SVHN | GTSRB | MNIST | DTD  | Avg Acc |
> | -------- | --------------------------------------- | ------ | ---- | -------- | ------- | ---- | ----- | ----- | ---- | ------- |
> | ViT-B/32 | Intersection of pair-wise trust regions | 62.3   | 58.9 | 71.7     | 81.3    | 79.5 | 71.6  | 96.7  | 55.0 | 72.1    |
> | ViT-B/32 | TATR                                    | 62.7   | 59.3 | 72.3     | 82.3    | 80.5 | 72.6  | 97.0  | 55.4 | 72.8    |
> | ViT-L/14 | Intersection of pair-wise trust regions | 74.0   | 83.5 | 87.3     | 93.5    | 87.2 | 86.2  | 98.9  | 66.7 | 84.7    |
> | ViT-L/14 | TATR                                    | 74.6   | 83.7 | 87.6     | 93.7    | 88.6 | 88.1  | 99.0  | 66.8 | 85.3    |

---

> ### Author Response · Authors · 2024-11-21
> **Response to vK8u Part 2**
>
> **Q: Is it always possible to find orthogonal components for task vectors?**
>
> A:  Yes with high probability. The orthogonal components defined in our work refer to the vector which contains elements of task vector with near-zero product to other task gradients, denoted as $\Delta_i^{\perp} = \Delta_i \odot 1_{\{\nabla_\theta L_j(\theta_{\text{pre}})\odot \Delta_i\approx 0\}}$. These components are not difficult to find, since the gradient $\nabla_\theta L_j(\theta_{\text{pre}})$ tends to be sparse, especially in neural networks such as transformers that leverage the attention mechanism or ReLU activation [1].
>
> Additionally, the table below reports the distribution of the absolute products between task vector elements to gradients across all task pairs, where the value of $n$-th element is calculated as $\sum_i \sum_{j, i\neq j}  \big| \nabla_\theta L_j( \theta_{\textnormal{pre}})[n] \cdot \Delta_i [n]  \big|$. As observed, the majority of values falls within the range of (0, 0.005]. In other words, most vector components are orthogonal components.
>
> | Range           | Number of components |
> | --------------- | -------------------- |
> | (0.000, 0.005]  | 113446706            |
> | (0.005,  0.010] | 1800                 |
> | (0.010, 0.015]  | 123                  |
> | (0.015,  0.020] | 37                   |
> | (0.020, 0.025]  | 17                   |
> | (0.025,  0.030] | 7                    |
> | (0.030, 0.035]  | 3                    |
> | (0.035,  0.040] | 2                    |
> | (0.040, 0.045]  | 1                    |
> | (0.045,  0.050] | 1                    |
> | (0.050, 0.055]  | 0                    |
> | (0.055,  0.060] | 1                    |
> | (0.060, 0.065]  | 1                    |
> | (0.065,  0.070] | 2                    |
> | (0.070, 0.075]  | 3                    |
> | (0.075,  0.080] | 1                    |
>
> [1] Li, Chunyuan, et al. "Measuring the Intrinsic Dimension of Objective Landscapes." *International Conference on Learning Representations*. 2018.
>
> **Q: Evaluation of heterogeneous task.**
>
> A: The experiment is currently in progress, and we are striving to finalize and release the results at the earliest opportunity. I appreciate your understanding.

---

> ### Author Response · Authors · 2024-11-29
> **Results of TATR with heterogeneous task**
>
> **Q: Evaluation of heterogeneous task.**
>
> A: Thank you for your patience. We followed your suggestion to perform experiments on PASCAL-Context. Specifically, we utilized the encoder of SAM with a ViT-B architecture as the pre-trained backbone, and fine-tuned it independently on four diverse tasks: semantic segmentation, human part segmentation, saliency estimation, and surface normal estimation.
>
> The following table exhibits the results, where TATR consistently improves the performance of TA-based methods.
>
> | Task | Semantic Segmentation  |  Human Part Segmentation | Saliency Estimation | Surface Normal Estimation |
> | --- | --- | --- | --- | --- |
> | Metric | mIoU (% $\uparrow$) | mIoU (% $\uparrow$) | mIoU (% $\uparrow$) | rmse ($\downarrow$) |
> | $\theta_{pre}$ | 34.43 | 51.65 | 61.23 | 20.44 |
> | TA | 35.55 | 56.38 | 61.24 | 20.54 |
> | TA + TATR | **41.38** | **57.12** | 62.60 | 20.05 |
> | Ties-Merging | 36.23 | 56.36 | 61.56 | 20.52 |
> | Ties-Merging + TATR | 40.24 | 57.00 | **62.67** | **20.02** |

---

> ### Author Response · Authors · 2024-12-01
> **Results of TATR with heterogeneous task Part2**
>
> **Q: Evaluation of heterogeneous task.**
>
> A: We also conducted experiments on vision-language tasks. We obtained the task vectors by finetuning the BLIP model [1] independently on three diverse vision-language datasets: TextVQA [2], ScienceQA [3], and SVR [4]. Among these, TextVQA focuses on answering questions by reading the text on the image. ScienceQA is about answering visual questions in the scientific domain, which involves knowledge recall and reasoning. SVR is about classifying textual descriptions regarding spatial relations in the image into true or false statements.
>
> More specifically, we finetuned the VQA version of BLIP for 6000 steps per task. The model architecture contains an image encoder, a text encoder, and a text decoder. All model weights were finetuned.
>
> The table below presents the results, demonstrating that TATR enhances the overall performance of TA-based methods.
>
> | Task | TextVQA | ScienceQA | SVR | Average |
> | --- | --- | --- | --- | --- |
> | Metric | Accuracy | Accuracy | Accuracy |  |
> | $θ_{pre}$ | 21.08 | 40.50 | 49.56 | 37.05 |
> | TA | 18.68 | 49.18 | 66.56 | 44.81 |
> | TA + TATR | **21.74** | 47.84 | **68.47** | 46.02 |
> | TiesMerging | 19.44 | **51.06** | 66.71 | 45.74 |
> | TiesMerging+TATR | 21.37 | 50.47 | 68.37 | **46.74** |
>
> [1] Li, Junnan, et al. "Blip: Bootstrapping language-image pre-training for unified vision-language understanding and generation." *International conference on machine learning*. 2022.
>
> [2] Singh, Amanpreet, et al. "Towards VQA models that can read." *Proceedings of the IEEE/CVF conference on computer vision and pattern recognition*. 2019.
>
> [3] Lu, Pan, et al. "Learn to explain: Multimodal reasoning via thought chains for science question answering." *Advances in Neural Information Processing Systems.* 2022.
>
> [4] Liu, Fangyu, Guy Emerson, and Nigel Collier. "Visual spatial reasoning." *Transactions of the Association for Computational Linguistics.* 2023.

---

### Author Response · Authors · 2024-11-21
**General Reply to All Reviewers**

We thank the reviewers for their critical assessment and constructive suggestions. We are delighted that they found our work to be well-motivated (YLeq), containing extensive experiments (YLeq, V85r), and well-written (vK8u, YLeq, H7en, V85r). They (vK8u, YLeq, H7en) agree that the paper is promising and practical. We will incorporate all feedback and additional experiments in the final version.

---

### Meta-Review · Area_Chair_ZCz8 · 2024-12-22

**Metareview:**

Summary: The paper introduces TATR, a method to address "knowledge conflicts" in multi-task model merging by defining a trust region for parameter updates that minimally affect task-specific losses. TATR aims to mitigate conflicts by merging parameters within this region, enhancing multi-task performance without additional training. The approach is shown to be effective across various datasets and can be integrated with existing methods.

Strengths:

TATR provides a clear motivation and a practical solution to the problem of knowledge conflicts in multi-task learning, with the potential to enhance the performance of existing task arithmetic methods.

The paper demonstrates TATR's effectiveness through extensive experiments on multiple datasets, showing improvements in multi-task performance.

Drawbacks:

The proposed method's reliance on orthogonal components for model merging may introduce task-irrelevant noise, which is not well-theoretically justified and could impact model performance.

There are concerns about the generalizability of TATR to more diverse and heterogeneous task settings, as the experiments are primarily conducted on relatively homogeneous classification tasks.

The performance of TATR is inconsistent across different datasets, with poor performance observed on the Cars dataset, raising questions about the method's robustness.

Given the mixed results and the concerns raised about the method's theoretical underpinnings and robustness, I must reject this work as it is a borderline paper that does not fully meet the acceptance criteria.

**Additional Comments On Reviewer Discussion:**

Concerns are not well-addressed.

---

### Decision · Program_Chairs · 2025-01-22

Reject